# PRIVATE WASSERSTEIN DISTANCE

## ABSTRACT

Wasserstein distance is a key metric for quantifying data divergence from a distributional perspective. However, its application in privacy-sensitive environments, where direct sharing of raw data is prohibited, presents significant challenges. Existing approaches, such as Differential Privacy and Federated Optimization, have been employed to estimate the Wasserstein distance under such constraints. However, these methods often fall short when both accuracy and security are required. In this study, we explore the inherent triangular properties within the Wasserstein space, leading to a novel solution named `TriangleWad`. This approach facilitates the fast computation of the Wasserstein distance between datasets stored across different entities, ensuring that raw data remain completely hidden. TriangleWad not only strengthens resistance to potential attacks but also preserves high estimation accuracy. Through extensive experiments across various tasks involving both image and text data, we demonstrate its superior performance and significant potential for real-world applications.

## 1 INTRODUCTION

Optimal Transport (OT) is one of the representative approaches that provides a geometric view that places a distance on the space of probability measures Villani et al. (2009). Specifically, it aims to find a coupling matrix that moves the source data to the target data with smallest cost, thereby inducing the Wasserstein distance, a metric used to measure the divergence between two distributions. Due to its favorable analytical properties, such as computational tractability and the ability to be computed from finite samples, the Wasserstein distance has been applied in various domains, including document similarity measurement Kusner et al. (2015), domain adaption Courty et al. (2016; 2017), geometric measurement between labelled data Alvarez-Melis & Fusi (2020), generative adversarial networks Arjovsky et al. (2017), dataset valuation and selection Just et al. (2023); Kang et al. (2024).

However, calculating the Wasserstein distance often requires access to raw data, which restricts its use in privacy-sensitive environments. In Federated Learning (FL), for instance, multiple parties collaboratively train a model without sharing raw data, while their data are usually non-independently and identically distributed (Non-IID) Li et al. (2022). In this context, the Wasserstein distance can be used to measure data heterogeneity, cluster clients with similar distributions, filter out out-of-distribution data, and ultimately improve FL model performance. However, since raw data cannot be accessed in the FL setting, direct computation of the Wasserstein distance becomes infeasible. Similarly, in a data marketplace, buyers seek to acquire training data from multiple sellers to build models for specific predictive tasks. However, sellers are often reluctant to grant access to their data prior to transactions due to the risk of data being copied, while buyers are hesitant to make purchases without first assessing the data's value, quality, and relevance Lu et al. (2024). In this case, a promising approach to aligning the interests of data sellers and buyers is to compute the Wasserstein distance between datasets in a privacy-preserving manner.

Recently, FedWad Rakotomamonjy et al. (2024) takes the first step to approximate the Wasserstein distance between two parties via triangle inequality. However, its applicability is limited to scenarios involving only two parties, making it unsuitable for data marketplaces with multiple data sellers, where the Wasserstein distance between aggregated training data (from multiple sellers) and validation data (held by the buyer) is required. Moreover, privacy concerns arise due to the shared information used to facilitate FedWad calculations, which unintentionally exposes raw images from both parties. By exploiting optimization conditions, it is even possible to reconstruct "clean' images from the shared

data. Privacy risks are even more pronounced when dealing with textual data, as shared information in the embedding space can reveal most of the original raw words. All of the aforementioned risks are undesirable for high-sensitive parties and make FedWad unacceptable in real-world applications. FedBary Li et al. (2024b) extends the previous work and addresses the task of noisy data detection based on shared information, but it suffers from an asymmetry in detection capabilities: only clients in FL or sellers in the data marketplace know exactly which data points are noisy, while the server in FL or data buyers lack this information. This asymmetry becomes particularly problematic when data sellers or clients are not trusted. Therefore, there is an urgent need for a privacy-enhanced approach to Wasserstein distance computation that ensures efficiency, accuracy, and symmetry in detection capabilities, especially in settings involving sensitive data and multiple parties.

This paper aims to develop a faster and more secure method for approximating the Wasserstein distance without sacrificing much accuracy. Our approach is based on geometric intuition derived from the intercept theorem associated with geodesics: by constructing two similar triangles, we establish a proportional relationship between their corresponding sides. This enables the direct approximation of the Wasserstein distance between two data distributions through the distance of their parallel segments. With our approach, accurate estimation can be achieved in just one round of interaction, significantly reducing computational costs. Moreover, as we reduce the interactions and change the optimization condition, our approach mitigates the privacy concerns associated with previous methods, which will be discussed in detail later. Thanks to its scalability, efficiency, and effectiveness, this solution addresses various real-world challenges. These include calculating client contributions in FL, performing clustering in FL, filtering out corrupt data points before training, assessing data relevance in data marketplaces, and any other privacy-sensitive contexts that require measuring distributional similarity.

**Our major contributions:** (1) We conduct a comprehensive theoretical analysis of geometric properties within the Wasserstein space, design a distributional attack against FedWad and introduce a novel approach, `TriangleWad`; (2) `TriangleWad` is simple, fast, accurate and enhances privacy. It also significantly improves the detection of noisy data from the server side in FL, better aligning with real-world requirements; (3) We conduct extensive experiments on both image and text datasets, covering a range of applications such as data evaluation, noisy data detection, and word movers distance, demonstrating its strong generalization capabilities.

## 2 PRELIMINARY AND RELATED WORK

### 2.1 RELATED WORK

**Private Wasserstein Distance** There are very few efforts to provide privacy guarantees for computing Wasserstein distance when raw data is forbidden to be shared. The first attempt is to apply Differential Privacy (DP) Lê Tien et al. (2019) with Johnson-Lindenstrauss transform. However, this approach is used for domain adaptation tasks, where only the source distribution is perturbed while the target distribution remains unchanged. Additionally, it does not have geometric property, and has inaccurate estimation empirically. The following work Rakotomamonjy & Liva (2021) considers DP for Sliced Wasserstein distance, and Jin & Chen (2022) uses DP for graph embeddings. Recently proposed FedWad Rakotomamonjy et al. (2024) develops a Federated way to approximate distance iteratively based on geodesics and interpolating measures, and FedBary Li et al. (2024b) extends this approach to approximate data valuation and Wasserstein barycenter, which could further be used for distributionally robust training Li et al. (2024a). It is worthy to note that one latest work Wasserstein Differential Privacy Yang et al. (2024) focuses computing privacy budgets through Wasserstein distance, which is not related to our applications.

**Data Evaluation in FL and Data Marketplace** Data quality valuation has gained more attentions in recent years since it has impact on the trained models and downstream tasks. Due to privacy issue, e.g. Federated Learning, only model gradients are shared for evaluation. Therefore, Shapley value (SV) Song et al. (2019); Jia et al. (2019); Liu et al. (2022); Xu et al. (2021a;b) is mainly used to measure client contributions as it provides marginal contribution score. Recently, based on Rakotomamonjy et al. (2024), FedBary Li et al. (2024b) uses Wasserstein distance to measure dataset divergence as the score of client contribution, and it leverages sensitivity analysis to further select valuable data points. In this paper, we focus on the horizontal FL, where clients' data shares the same feature space. Data evaluation with privacy guarantees is also applied in data marketplaces,

where evaluation must be conducted before granting data access. Recently, DAVED Lu et al. (2024) proposed a federated approach to the data selection problem, inspired by linear experimental design, which achieves lower prediction error without requiring labeled validation data.

## 2.2 Optimal Transport and Wasserstein Distance

**Definition 1** *(Wasserstein distance) The $p$-Wasserstein distance between measures $\mu$ and $\nu$ is*

$$\mathcal{W}_p(\mu,\nu) = \Big( \inf_{\pi \in \Pi(\mu,\nu)} \int_{\mathcal{X} \times \mathcal{X}} d^p(x,x') d\pi(x,x') \Big)^{1/p}, \tag{1}$$

*where $d(x,x')$ is the pairwise distance metric, e.g. Euclidean distance. $\pi \in \Pi(\mu,\nu)$ is the joint distribution of $\mu$ and $\nu$, and any $\pi$ attains such minimum is considered as an optimal transport plan.*

In the discrete space, the two marginal measures are denoted as $\mu = \sum_{i=1}^{m} a_i \delta_{x_i}, \nu = \sum_{j=1}^{n} b_j \delta_{x'_j}$ ,where $\delta_{x_i}$ is the dirac function at location $x_i \in \mathbb{R}^d$, and $a_i$ and $b_i$ are probability masses associated to the $i$- sample and belong to the probability simplex, $\sum_{i=1}^{m} a_i = \sum_{j=1}^{n} b_j = 1$. Therefore, the Monge problem seeks a map that associates to each point in $x_i$, a single point $x'_j$ and which must push the mass of $\mu$ toward the mass of $\nu$. However, when $m \neq n$, the Monge maps may not exist between a discrete measure to another, especially when the target measure has larger support size of the source measure. Therefore, we consider the Kantorovich's relaxed formulation, which allows *mass splitting* from a source toward several targets. The Kantorovich's optimal transport problem is

$$\mathcal{W}_p(\mu,\nu) = (\min_{\mathbf{P} \in \Pi(\mu,\nu)} \langle \mathbf{C}, \mathbf{P} \rangle)^{1/p} \tag{2}$$

where $\mathbf{C} \doteq (d_X^p(x_i, x'_j)) \in \mathbb{R}^{m \times n}$ is the matrix of all pairwise costs, and $\Pi(\mu,\nu) = \{\mathbf{P} \in \mathbb{R}_+^{m \times n} | \mathbf{P}\mathbf{1}_m = \mu, \mathbf{P}^\top \mathbf{1}_n = \nu\}$ is the set of all transportation couplings.

## 2.3 Wasserstein Geodesics and Interpolaing Measure

**Definition 2** *(Wasserstein Geodesics, Interpolating measure Rakotomamonjy et al. (2024); Ambrosio et al. (2005)) Denote $\mu, \nu \in \mathcal{P}_p(\mathcal{X})$ with $\mathcal{X} \subseteq \mathbb{R}^d$ compact, convex and equipped with $\mathcal{W}_p$. Let $\pi \in \Pi(\mu,\nu)$ be an optimal transport plan. For $t \in [0,1]$, define $\eta(t) = ((1-t)x + tx')_{\#}\pi$, $x \sim \mu, x' \sim \nu$, thus $\eta(t)$ is the push-forward measure under the map $\pi$. Then, the curve $\bar{\mu} = (\eta(t))_{t \in [0,1]}$ is a constant speed geodesic between $\mu$ and $\nu$, also called a Wasserstein geodesics between $\mu$ and $\nu$. Any point $\eta(t)$ on $\bar{\mu}$ is an interpolating measure between distribution $\mu$ and $\nu$, as expected*

$$\mathcal{W}_p(\mu,\nu) = \mathcal{W}_p(\mu,\eta(t)) + \mathcal{W}_p(\eta(t),\nu). \tag{3}$$

In the discrete setup, denoting $\mathbf{P}^\star$ a solution of equation 2, an interpolating measure is obtained as

$$\eta(t) = \sum_{i,j}^{m,n} \mathbf{P}^\star_{i,j} \delta_{(1-t)x_i + tx'_j}, \tag{4}$$

where $\mathbf{P}^\star_{i,j}$ is the $(i,j)$-th entry of $\mathbf{P}^\star$, and the maximum number of non-zero elements of $\mathbf{P}$ is $n + m - 1$. Rakotomamonjy et al. (2024) proposes to use the barycentric mapping to approximate the interpolating measure as

$$\eta(t) = \frac{1}{m} \sum_{i=1}^{m} \delta_{(1-t)x_i + tm(\mathbf{P}^\star \mathbf{x}^\nu)_i} \tag{5}$$

where $x_i$ is $i$-th support from $\mu$, $\mathbf{x}^\nu$ is the matrix of $\nu$. When $m = n$, equation 4 and equation 5 are exactly equivalent. In both equation 4 and 5, the parameter $t$ is defined as *push-forward parameter*, which controls how much we could push forward the source distribution $\mu$ to the target distribution $\nu$, and construct the interpolating measure $\eta(t)$.

## 3 METHODOLOGY

### 3.1 PROBLEM STATEMENT

Our goal is to compute the Wasserstein distance among different datasets distributed on separate parties, with the constraint that raw data is not shared. Without loss of generality, we start with the case to calculate the Wasserstein distance between two measures, which can be easily extended to measuring the divergence among multiple measures. We consider the 2-Wasserstein distance $\mathcal{W}_2(\cdot, \cdot)$ in this paper, while our proposed approach can be generalized to other $p$ cases.

### 3.2 INTUITION AND MOTIVATION

Based on equation 3, FedWad Rakotomamonjy et al. (2024) proposes a Federated manner to approximate the interpolating measure $\xi$ between $\mu$ and $\nu$, and obtain the Wasserstein distance $\hat{\mathcal{W}}_2(\mu, \nu)$ via $\mathcal{W}_2(\mu, \xi) + \mathcal{W}_2(\xi, \nu)$. However, based on equation 4 and equation 5, directly calculating the interpolating measure $\xi$ needs to access to raw data from both sides. Therefore, two additional measures $\eta_\mu$ and $\eta_\nu$ are introduced to approximate $\xi$. The proposed approach decomposes the Wasserstein distance $\mathcal{W}_2(\mu, \nu)$ into 4 parts as follows, and the right-hand side provides an upper bound of the exact distance,

$$\mathcal{W}_2(\mu, \nu) \leq \hat{\mathcal{W}}_2^{(k)}(\mu, \nu) = \mathcal{W}_2(\mu, \eta_\mu^{(k)}) + \mathcal{W}_2(\eta_\mu^{(k)}, \xi^{(k-1)}) + \mathcal{W}_2(\xi^{(k-1)}, \eta_\nu^{(k)}) + \mathcal{W}_2(\eta_\nu^{(k)}, \nu). \quad (6)$$

Specifically, $\xi^{(0)}$ is randomly initialized and shared with both parties. For every round $k$, each party calculates the interpolating measure $\eta_\mu^{(k)}/\eta_\nu^{(k)}$ between $\mu/\nu$ and $\xi^{(k-1)}$, respectively. Then $\eta_\mu^{(k)}$ and $\eta_\nu^{(k)}$ are shared to optimize a new $\xi^{(k)}$, which is an interpolating measure between $\eta_\mu^{(k)}$ and $\eta_\nu^{(k)}$. With iterative optimizations, all $\eta_\mu^{(K)}, \xi^{(K)}, \eta_\nu^{(K)}$ will converge to interpolating measures between $\mu$ and $\nu$ at $K$-th round, then equation 6 will become an equation, such that $\mathcal{W}_2(\mu, \nu) = \hat{\mathcal{W}}_2^{(k)}(\mu, \nu)$.

During above iterations, public information contains a set of $\{\eta_\mu^{(k)}, \eta_\nu^{(k)}, \xi^{(k)}\}_{k=0}^K$, and Wasserstein distances $\mathcal{W}_2(\mu, \xi^{(K)})$ and $\mathcal{W}_2(\xi^{(K)}, \nu)$ need be shared. Private information consists of the OT plans between $\mu$ and $\xi^{(k)}$, OT plans between $\xi^{(k)}$ and $\nu$, and parameters $t$ for constructing the interpolating measure. The privacy advantage lies in keeping the OT plans and $t$ being private. However, we identify a potential privacy risk when equation 6 holds as an equality. Even without access to the informative elements (OT plans and $t$), an attacker could still infer raw data. Firstly, we observe that the interpolating measure $\xi^{(K)}$ can significantly leak raw data when computing Wasserstein distance for textual structured data. For instance, retrieving the top-1 similar words within the embedding space of $\xi^{(K)}$ (Figure 3) reveals that most of these words originate directly from the raw text of both parties. Secondly, there is a potential distributional attack in which an attacker could leverage the available information to construct the approximation that has a very small Wasserstein distance from the raw data. This is undesirable to some high-sensitive parties such as hospitals. Suppose the attacker holds $\mu$, and he wants to infer information of $\nu$ from the other side. Available information for this attacker is: $\mathcal{W}_2(\mu, \nu)$, $\mathcal{W}_2(\mu, \xi^{(K)})$, $\mu$, $\eta_\mu^{(K)}$, and $\xi^{(K)}$. Therefore, the intuition of the proposed attack is straightforward: Two Wasserstein balls $\mathcal{B}(\mu, \mathcal{W}_2(\mu, \xi^{(K)}))$ and $\mathcal{B}(\mu, \mathcal{W}_2(\mu, \nu))$, along with the condition that $\mu, \xi^{(K)}, \nu$ lie on the same geodesics, could uniquely determine the distribution of $\nu$. The attacker could initialize a learnable attack data matrix $\hat{\nu}$, and computes the distance $\mathcal{W}_2(\hat{\nu}, \mu)$ and $\mathcal{W}_2(\hat{\nu}, \xi^{(K)})$. In empirical experiments, we relax the constraint that $\hat{\nu}$ is on the same geodesics with $\mu$ and $\nu$, and only meet two conditions: $\mathcal{W}_2(\mu, \hat{\nu}) = \mathcal{W}_2(\mu, \nu)$ and $\mathcal{W}_2(\hat{\nu}, \xi^{(K)}) = \mathcal{W}_2(\nu, \xi^{(K)}) = \mathcal{W}_2(\nu, \mu) - \mathcal{W}_2(\xi^{(K)}, \mu)$. Then we find we could get $\hat{\nu}$ such that $\mathcal{W}_2(\hat{\nu}, \nu) \simeq 0$, which means the attack data and raw data are distributional identical. Empirical results are shown in Appendix D.1.

### 3.3 PROPOSED SOLUTION

From the previous discussion, we observe a trade-off between privacy and accuracy: performing exact calculations constructs Wasserstein balls, which provide geometric information that could reveal the distribution of the raw data. Therefore, our proposed solution is to avoid constructing any

| | FedWad | Ours |
|---|---|---|
| Calculation | $\mathcal{W}(\mu,\eta_\mu) + \mathcal{W}(\eta_\mu^{(K)} + \xi^{(K)})$ $+\mathcal{W}(\xi^{(K)},\eta_\nu^{(K)}) + \mathcal{W}(\eta_\nu^{(K)},v)$ | $\frac{1}{(1-t)}\mathcal{W}(\eta_\mu,\eta_\nu)$ |
| Conditions | $\eta_\mu^{(K)},\eta_\nu^{(K)},\xi^{(K)} \in IM(\mu,\nu)$ | $\eta_\mu \in IM(\mu,\gamma)$ $\eta_\nu \in IM(\nu,\gamma)$ |
| Error Bound | Convergence Analysis | $\sigma_\gamma^2(p_u - p_v)^2$ |
| Iterations | $K$ rounds until converge | one round |
| Common Information | $\xi^{(k)},\eta_\mu^{(k)},\eta_\nu^{(k)}, k \in [1,K]$ $\mathcal{W}(u,\xi^{(K)}),\mathcal{W}(\xi^{(K)},v)$ | $\gamma,\eta_\mu,\eta_\nu$ $\mathcal{W}(\eta_\mu,\eta_\nu)$ |
| Private Information | $\pi(\mu^{(k)},\xi^{(k-1)}),\pi(\nu^{(k)},\xi^{(k-1)}),t^{(k)}$ | $\pi(\mu,\gamma),\pi(\nu,\gamma),t$ |
| Distributional Attack | ✓ | × |

Figure 1: **Technical Comparison**: In previous work Rakotomamonjy et al. (2024), two Wasserstein balls $\mathcal{B}(\mu,\mathcal{W}_2(\mu,\xi))$ and $\mathcal{B}(\mu,\mathcal{W}_2(\mu,\nu))$, along with the condition that $\mu,\xi,\nu$ lie on the same geodesics, could uniquely determine the distribution of $\nu$. TriangleWad does not have such an interpolating measure between $\mu$ and $\nu$. Simultaneously, $\mathcal{W}_2(\nu,\gamma),\mathcal{W}_2(\nu,\eta_\mu),\mathcal{W}_2(\eta_\nu,\gamma)$ are private information. $IM(a,b)$ represents the interpolating measure between $a$ and $b$

interpolating measures between raw distributions and to minimize interactions as much as possible. The technical comparison is shown in Figure 1. Suppose $\mu \in \mathbb{R}^{m \times d}, \nu \in \mathbb{R}^{k \times d}$, where $\mu$ and $\nu$ are raw data held by two separate parties, $\gamma \in \mathbb{R}^{n \times d} \sim \mathcal{N}(m_\gamma, \sigma_\gamma^2)$ is a randomly initialized gaussian measure. If $\eta_\mu(t)$ is an interpolating measure between $\mu$ and $\gamma$, $\eta_\nu(t)$ is an interpolating measure between $\nu$ and $\gamma$, we state there is a proportional relationship between $\mathcal{W}_2(\eta_\mu, \eta_\nu)$ and $\mathcal{W}_2(\mu, \nu)$ as

$$\mathcal{W}_2(\mu,\nu) \leq \hat{\mathcal{W}}_2(\mu,\nu) = \frac{1}{1-t}\mathcal{W}_2(\eta_\mu,\eta_\nu). \tag{7}$$

The geometric intuition behind is the intercept theorem: if $\eta_\mu$ is on the segment $[\gamma,\mu]$, $\eta_\nu$ is on the segment $[\gamma,\nu]$, given the segment $[\eta_\mu,\eta_\nu]$ is parallel to the segment $[\mu,\nu]$, there is a proportional relationship between $\mathcal{W}_2(\eta_\mu,\eta_\nu)$ and $\mathcal{W}_2(\mu,\nu)$. We follow the same barycentric mapping in equation 5 and analyze the error bound between $\hat{\mathcal{W}}_2(\mu,\nu)$ and $\mathcal{W}_2(\mu,\nu)$ as follows,

**Theorem 1** *Suppose $\gamma \in \mathbb{R}^{k \times d} \sim \mathcal{N}(\mu_\gamma,\sigma_\gamma^2)$. Let $\pi^\star(\mu,\gamma) \in \mathbb{R}^{m \times k}$ be the OT plan between $\mu$ and $\gamma$, $\pi^\star(\nu,\gamma) \in \mathbb{R}^{n \times k}$ be the OT plan between $\nu$ and $\gamma$. If $\eta_\mu$ and $\eta_\nu$ are approximated by Eq. equation 8 as*

$$\eta_\mu(t) = \frac{1}{m}\sum_{i=1}^m \delta_{(1-t)\mu_i + mt[\pi^\star(\mu,\gamma)\gamma]_i} \quad \eta_\nu(s) = \frac{1}{n}\sum_{i=1}^n \delta_{(1-s)\nu_i + ns[\pi^\star(\nu,\gamma)\gamma]_i}. \tag{8}$$

*with the condition that both measures have the same push parameters, e.g. $t = s$, then the approximation error $|\hat{\mathcal{W}}^2(\mu,\nu) - \mathcal{W}_2^2(\mu,\nu)|$ is bounded by $\mathcal{O}(C\sigma_\gamma^2)$, where $C << 1$, and it has a negative relationship with $k$: the data size of $\gamma$.*

The proof is shown in Appendix. Additionally, we provide the proof that for the general $\mathcal{W}_p$, the approximation error $|\hat{\mathcal{W}}_p^p(\mu,\nu) - \mathcal{W}_p^p(\mu,\nu)|$ is bounded by the $p$-th sample moments of $\gamma$, which also aligns with the conclusion in Theorem 1. $t$ and $s$ are parameters to control how much we could push forward the raw data to the target distribution $\gamma$ and construct the interpolating measure. This theorem tells that if both sides calculate the interpolating measures between their own data and $\gamma$ with the same push-forward parameter $t$, then they can easily approximate the Wasserstein distance with trivial errors. Furthermore, based on the proof of the Theorem 1, there are some special cases that the approximated Wasserstein distance is the same as the true Wasserstein distance, such that equation 7 becomes an equation.

**Corollary 1** *If one of the following condition holds: (1) $\sigma_\gamma = 0$; (2) $k = 1$; (3) $k \to \infty$; (4) $\mu$ and $\nu$ are Gaussian distributions with the same covariance matrix or $m = n$, then the approximation value $\hat{\mathcal{W}}_p(\mu,\nu)$ is exactly the true distance $\mathcal{W}_p(\mu,\nu)$.*

**Corollary 2** *Suppose $\gamma \sim \mathcal{N}(\bar{\gamma},\sigma_\gamma^2)$, then each element of $\eta_\mu$ is obtained by the linear transformation with the Gaussian distribution $\mathcal{N}(\bar{\gamma},\sigma^2(\pi^\star(\mu,\gamma)\gamma))$ as follows,*

$$\eta_\mu(t) = \sum_i \delta_{(1-t)\mu_i + t[\bar{\gamma} + \sigma(\pi^\star(\mu,\gamma)\gamma)z_i]}. \tag{9}$$

where $z_i \sim \mathcal{N}(0, 1)$. If $k \to \infty$, then $\sigma(\pi^\star(\mu, \gamma)\gamma) \to 0$.

Corollary 2 demonstrates that constructing the interpolating measure is equivalent to adding general perturbations to the raw data. The noise level is influenced by $t$, which controls the weights, as well as by the randomness introduced through $\pi^\star(\mu, \gamma)\gamma$. Both Theorem 1 and Corollary 9 suggest scaling up $\gamma$ and setting a larger variance for $\gamma$ can be strategic choices for enhancing randomness. Scaling up $\gamma$ increases the complexity of the OT plan, and setting a larger variance for $\gamma$ directly boosts randomness. However, these two strategies have conflicting effects on $\sigma(\pi^\star(\mu, \gamma)\gamma)$, necessitating careful tuning to balance utility and privacy. In practice, we could set $k \simeq \min\{m, n\}$.

**Remark 1** *Advantages of approximating the interpolating measure: The OT plan between $\mu/\nu$ and $\gamma$ has at most $(m + k - 1)/(n + k - 1)$ non-zero elements, when $m \neq n \neq k$. If we use the exact calculation as in equation 4, the larger size of the interpolating measures $\eta_\mu$ and $\eta_\nu$ will potentially lead to significant computational overhead. However, with barycentric mapping as in equation 5, we can ensure that the size of the interpolating measures $\eta_\mu$ and $\eta_\nu$ remains consistent with $\mu$ and $\nu$, respectively, which helps reduce computational costs, as discussed in Sec 4.1. Additionally, from Corollary 2, we observe that the interpolating measure is equivalent to a linear transformation of the raw data, which is useful for detecting noisy data points. This will be further discussed in Sec 3.5.*

### 3.4 Approximate Wasserstein Distance with unknown $t$

Theorem 1 states that the approximation error is minimized when the interpolating measures $\eta_\mu(t)$ and $\eta_\nu(t)$ are calculated using the same $t$ via equation 8, implying that the value of $t$ should be public information. As discussed in Sec 3.2, OT plans and push-forward parameters are key elements for reconstructing raw data and should remain private. While making $t$ public might seem to contradict privacy guarantees, we argue that the OT plan is the most critical component for reconstructing raw data, and it is impossible to reconstruct raw data without access to this information, which will be discussed in detail in Sec 4.2.1. However, we find when computing Wasserstein distance among multiple data distributions, there is a solution to hide such push-forward parameter. Before explaining the calculation procedure, we present the following theorem,

**Theorem 2** *Given a fixed measure $\eta_\mu(t_0)$, which is the interpolating measure between $\mu$ and $\gamma$ at a fixed value $t_0 \in (0, 1)$. Let $\eta_\nu(s)$ be the interpolating measure between $\nu$ and $\gamma$ with $\forall s \in (0, 1)$. Then the 2-Wasserstein distance $\mathcal{W}_2(\eta_\mu(t), \eta_\nu(s))$ is a quadratic function with respective to the value of s, such that*

$$\mathcal{W}_2^2(\eta_\mu(t_0), \eta_\nu(s)) = f(s) = a_2 s^2 + a_1 s + a_0, \tag{10}$$

*where $a_2, a_1, a_0$ are constant coefficients.*

The proof is shown in Appendix B. Specifically, the party A calculates the measure $\eta_\mu(t_0)$, where $t_0$ is his private information. Then party A shares $\eta_\mu(t_0)$ with the party B, and requires the set of tuples $\{s_j, \mathcal{W}_2(\eta_\nu(s_j), \eta_\mu(t_0))\}_{j=1}^{B_s}$, where $s_j \in (0, 1)$, $B_s$ is the sampling budget and $\mathcal{W}_2(\eta_\nu(s_j), \eta_\mu(t_0))$ is calculated by the party B. Then the party A could fit an estimator function $f(s) = \mathcal{W}_2(\eta_\mu(t_0), \eta_\nu(s))$ based on equation 11, and calculate $\hat{\mathcal{W}}^2(\mu, \nu)$ with $\frac{1}{1-t_0} f(t_0)$.

$$(\hat{a}_0, \hat{a}_1, \hat{a}_2) = \arg\min_{a_0, a_1, a_2} \sum_{j=1}^{B_s} \left( \hat{\mathcal{W}}^2(\eta_\nu(s_j), \eta_\mu(t_0)) - \mathcal{W}_2^2(\eta_\nu(s_j), \eta_\mu(t_0)) \right)^2. \tag{11}$$

In practice, we opt for the choice of $s_j \in \{\frac{1}{4}, \frac{1}{2}, \frac{3}{4}\}$, which is enough to provide accurate estimations. Once the parameters are learned, the distance predictor can be used to predict the Wasserstein distance by plugging true push-forward parameter $t_0$ as input to the predictor.

The above procedures could be applied in the data marketplace, when there are multiple data sources $\{\nu_i\}_{i=1}^N$, and the data buyer wants to know the Wasserstein distance between the aggregated data $\sum_{i=1}^N \nu_i$ and his own validation set $\mu$, e.g. $\mathcal{W}_2(\sum_{i=1}^N \nu_i, \mu)$. Follow the similar procedure, a global shared random distribution $\gamma$ is initialized. The buyer calculates $\eta_\mu(t_0)$ and sends it to each data seller without sharing the value of $t_0$. Then the $i$-th data seller calculates the cost matrix $\mathbf{C}_i(s_j) = \mathbf{C}_i(\eta_{\nu_i}(s_j), \eta_\mu(t_0))$, which represents the point-wise euclidean distance between the interpolating measure $\eta_{\nu_i}(s_j)$ and $\eta_\mu(t_0)$, where $s_j$ is the sampling ratio requested by the data buyer. Then the

concatenated cost matrix $\mathbf{C}(s_j) = [\mathbf{C}_i(s_j), \cdots, \mathbf{C}_N(s_j)]^T$ is utilized to optimize the OT problem, and calculate the 2-Wasserstein distance $\mathcal{W}_2^2(\sum_{i=1}^N \eta_{\nu_i}(s_j), \eta_\mu(t_0)) = \min_{\mathbf{P}} \langle \mathbf{C}(s_j), \mathbf{P} \rangle$. Finally the set $\{s_j, \mathcal{W}_2^2(\sum_{i=1}^N \eta_{\nu_i}(s_j), \eta_\mu(t_0))\}_{j=1}^{B_s}$ is used to approximate the parameters in equation 11, and the buyer could calculate the Wasserstein distance by putting the true value of $t_0$.

## 3.5 BROADER APPLICATIONS

We will explain how TriangleWad can be extended to various applications with minor modifications, which are useful for domain adaptation and data evaluation in privacy setting.

**Wasserstein Distance between labeled dataset** OOTD Alvarez-Melis & Fusi (2020) introduces an effective way to augment data representations for calculating Wasserstein distance with labeled data. It leverages the point-wise notion $z = (x, y) \in \mathcal{X} \times \mathcal{Y}$ for measurements

$$
\begin{aligned}
d(z, z') = d((x,y),(x',y')) &\triangleq \left(d(x,x') + \mathcal{W}_2^2(\alpha_y, \alpha_{y'})\right)^{1/2}, \\
\mathcal{W}_2^2(\alpha_y, \alpha_{y'}) &= ||m_y - m_{y'}||_2^2 + ||\Sigma_y - \Sigma_{y'}||_2^2,
\end{aligned}
\tag{12}
$$

where $\alpha_{y'}$ is conditional feature distribution $P(\mathbf{x}|Y = y') \sim \mathcal{N}(m_{y'}, \Sigma_{y'}^2)$. However, the calculation of the interpolating measure requires the vectorial representation, which means the point-wise cost matrix could not be directly applied for our setting. For our extension, we follow the similar way in Rakotomamonjy et al. (2024), to incorporate the label information by constructing the augmented representation as $\mathbf{X} := [\mathbf{x}; m_y; \text{vec}(\Sigma_y^{1/2})]$. Therefore, when conducting approximations, all labeled datasets should be pre-processed into such form and the random initialisation of $\gamma$ follows the same dimension.

**Detecting valuable or noisy data points** Beyond calculating the Wasserstein distance between datasets, we could evaluate the "contribution score" of individual data point, to identify the valuable or noisy subsets. We take advantage of characteristics that the duality of the optimal transport problem is linear, and conduct the sensitivity analysis as in Just et al. (2023); Li et al. (2024b), to assign the score to individual data point. We use the interpolating measures $\eta_\mu$ and $\eta_\nu$ to conduct the evaluation, as a noisy data point in the raw data should also be the distributional outlier in the transformed form. The duality problem is $\mathcal{W}_2(\eta_\mu, \eta_\nu) = \max_{(f,g) \in C^0(\mathcal{Z})^2} \langle f, \eta_\mu \rangle + \langle g, \eta_\nu \rangle$, where $C^0(\mathcal{Z})$ is the set of all continuous functions, $f \in \mathbb{R}^{m \times 1}$ and $g \in \mathbb{R}^{n \times 1}$ are the dual variables. Then the constructed *gradient score* is as follows

$$
s_l = \frac{\partial \mathcal{W}_2(\eta_\mu, \eta_\nu)}{\partial \eta_\mu(z_l)} = f_l^\star - \sum_{j \in \{1, \cdots, m\} \setminus l} \frac{f_j^\star}{m-1},
\tag{13}
$$

which represents the rate of change in $\mathcal{W}_2(\eta_\mu, \eta_\nu)$ w.r.t. the given data point $z_l$ in $\eta_\mu$, likewise for $\eta_\nu$. The interpretation of the value $s_l$ is: the data point with the positive/negative sign of the score causes $\mathcal{W}_2(\eta_\mu, \eta_\nu)$ to increase/decrease, which is considered noisy/valuable. This score suggests removing data points with large positive gradient could help to match the target distribution. Li et al. (2024b) discovered the detection capability is unsymmetrical. In TriangleWad, introducing $\eta_\mu$ and $\eta_\nu$ engenders symmetrical capabilities in identifying noisy data, facilitating recognition by both the client and Server. This enhancement aligns more closely with real-world scenarios: Server can select valuable data points or identify potential attacks from clients.

## 4 THEORETICAL ANALYSIS

### 4.1 COMPLEXITY ANALYSIS

We conduct similar complexity analysis as in Rakotomamonjy et al. (2024). The communication cost involves the transfer of $\gamma$, $\eta_\mu$ and $\eta_\nu$. If the size of the data dimension is $d$, then the communication cost is $\mathcal{O}((k + m + n)d)$. As for the computational complexity of interpolating measures and Wasserstein distance, an appropriate choice of the support size of $\gamma$ is necessary. If we choose for the exact calculations, considering that $\mu$ and $\nu$ are discrete measures, $\eta_\mu$ and $\eta_\nu$ are supported on at most $m + k - 1$ and $n + k - 1$ respectively based on equation 4. Then for computing $\mathcal{W}_2(\eta_\mu, \eta_\nu)$, there are $(m + n + 2k - 2)$ non-negative elements in the OT plan, it might yield the computational overhead when all $n, m, S$ are large. Therefore, we opt for the choice of a smaller support size of $\gamma$, such as

$k = \min\{m, n\}$. Furthermore, with the barycentric mapping as in equation 5, we can guarantee that the support size of $\eta_\mu$ and $\eta_\nu$ are always $m$ and $n$ respectively. Therefore, for computing $\mathcal{W}_2(\eta_\mu, \eta_\nu)$, we can guarantee the computational complexity as $\mathcal{O}((n + m)nm log(n + m))$.

## 4.2 PRIVACY ANALYSIS

In this section, we will discuss two privacy benefits of our proposed approach.

**(1) Sec 4.2.1: Attackers lack important pieces of information to infer raw data:** Attackers can only infer raw data $\mu$ when they know $\eta_\mu(t_0)$, the OT plan $\pi(\mu, \gamma)$ and the value of push-forward parameter $t_0$, while the later two terms are kept private. And approximating the OT plan is an NP-hard problem; Compared to the previous approach, our approach involves only one round of interaction, which limits the available common information and hinders any attempts at approximation.

**(2) Sec 4.2.2:Setting a large $t_0$ could help protect privacy**: from a geometric perspective, it controls how much the interpolating measure is pushed closer to random Gaussian noise. From a statistical perspective, it introduces more substantial noise to the raw data. Consequently, a larger $t_0$ results in a greater Wasserstein distance between $\eta_\mu(t_0)$ and $\mu$, indicating higher dissimilarity between them.

### 4.2.1 DEFENSE TO ATTACKS

Traditional general attacks cannot be directly applied in our setting, as our approach does not involve any model training, and the shared information is insufficient to train a model. We consider two types of attack tailored to this research from both geometric and statistical views. Suppose an attacker tries to infer $\hat{\mu}$ based on available information, there are two potential attacks:

(1) *Distributional attack:* Whether $\mathcal{W}_2(\hat{\mu}, \mu) < \epsilon$ for a very small $\epsilon$?
(2) *Reconstruction attack:* Whether $\|\hat{\mu} - \mu\|^2 < \epsilon$ for a very small $\epsilon$?

The first attack is the distributional attack designed for FedWad. TriangleWad does not calculate any interpolating measure between $\mu$ and $\nu$, so the attacker can not use available information to identify the distribution of the raw data. For the second attack, the attacker knows the structure of equation 5, $\eta_\mu, \gamma$. The attacker might approximate $\hat{\mu} = \frac{1}{1-t}(\eta_\mu - t\gamma)$ while the groundtruth is $\frac{1}{1-t_0}(\eta_\mu - t_0\pi(\mu, \gamma)\gamma)$. In the worst case that $t_0$ becomes public information, it is also challenging for the attacker to reconstruct raw data, as private information $\pi(\mu, \gamma)$ has $m + k - 1$ non-zero elements with non-identical value, which is impossible to exactly approximate. Therefore, without knowing the exact value of both $t_0$ and $\pi(\mu, \gamma)$, $\hat{\mu}$ and $\mu$ will have a significant gap in both the Euclidean distance and Wasserstein distance, making the attack fail. We visualize this result in Figure 2 (lower right panel) in the experiments, and find each element in $\eta_\mu$ (Local IM) and $\hat{\mu}$ (Attack) are uninformative.

### 4.2.2 QUANTIFY THE DIFFERENCE BETWEEN $\mu$ AND $\eta_\mu(t)$

We have proved that the proposed approach preserves the procedure of normal perturbations with some randomness in Corollary 2. The following theorem helps to quantify the distance between the interpolating measure and the raw data.

**Theorem 3** *The 2-Wasserstein distance between raw data and the interpolating measure is proportional to the 2-Wasserstein distance between raw data and the random noises as*

$$\mathcal{W}_2(\mu, \eta_\mu(t)) = t\mathcal{W}_2(\mu, \gamma). \tag{14}$$

The proof is shown in Appendix. This result implies that we can set a larger $t \in (0, 1)$ to increase the dissimilarities between $\eta_\mu(t)$ and $\mu$ in Wasserstein space, thereby protecting privacy without sacrificing utility. The empirical result is shown in Appendix F.2.

## 5 EXPERIMENTS

We conduct experiments on both image and text datasets to demonstrate the efficiency and effectiveness of TriangleWad across multiple tasks. For the quantitative analysis, we expect TriangleWad to provide accurate estimations with reduced computational time. For the qualitative analysis, we

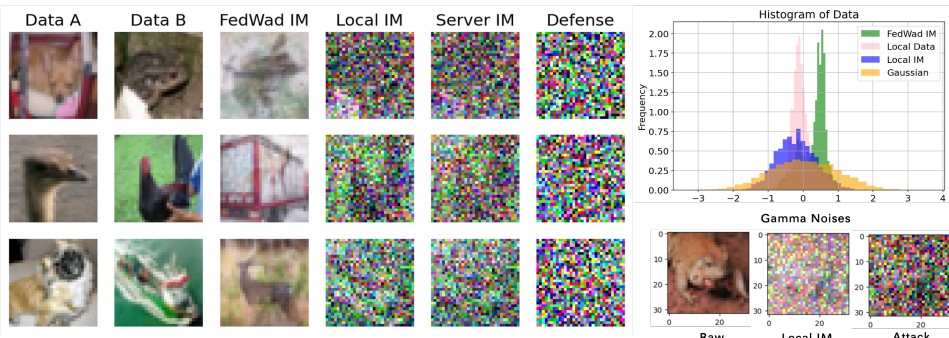

Figure 2: Qualitative Visualizations with Gaussian Noises: the global interpolating measure $\gamma$ in FedWad (FedWad IM) is visually informative, while in our approach, all interpolating measures (Local IM and Server IM) and $\gamma$ (Defense) are visually noisy. In addition, the statistic plot in the right side shows Local IM indeed follows the Gaussian distribution while FedWad IM is similar to raw data.

anticipate that the shared measure will reveal minimal information from the raw data: for image data, visual elements should be unrecognizable, and for text data, fewer raw words will be retrieved. Additional applications with empirical results are provided in the Appendix.

## 5.1 QUANTITATIVE AND QUALITATIVE ANALYSIS FOR IMAGE DATA

We employ CIFAR10, Fashion, and MNIST datasets as case studies to provide both quantitative and qualitative analyses among DirectWad, FedWad, and TriangleWad. DirectWad calculates Wasserstein distance with raw data directly, which is our ground truth. We will compare FedWad and TriangleWad in terms of their approximation differences from the ground truth and the average computation time and the results are summarized in Table 1. For data processing, we randomly select subsets $\mu$ and $\nu$ with equal sizes (100/500/1000), and their distributions do not necessarily to be identical. $D_1^{\text{noise}}$ and $D_2^{\text{noise}}$ are derived from the clean data $\mu$, with the former containing 20 noisy data points and the latter containing 50 noisy data points. The noisy type is to add the Gaussian noise in the feature space. For fair comparisons, we set $\gamma = \xi^{(0)} \sim \mathcal{N}(0, 1)$ for TriangleWad and FedWad, and the iteration epoch is set as 30 for FedWad because the optimization round does not affect the distance significantly when attaining the local convergence Rakotomamonjy et al. (2024). DirectWad provides the ground truth distance. FedWad is our baseline, using the triangle inequality to approximate the distance. The average gap refers to the distance gap with DirectWad, while the average time represents the computational cost. Therefore, TriangleWad provides competitive approximation accuracy with less computational time compared to FedWad. These findings emphasize the efficiency of our approach without compromising estimation precisions. The qualitative analysis aims to demonstrate the privacy guarantee, and we visualize the results in Figure 2. The left panel illustrates the CIFAR10 data distributed in parties A and B, $\gamma$ of FedWad between A and B (FedWad IM), $\eta_\mu$ and $\eta_\nu$ of TriangleWad (Local IM, Server IM), and randomly constructed $\gamma$ (Defense). Both Local IM and Server IM do not reveal any information about the data. From FedWad IM, we can identify the class of each image. In the right-side histogram plot, we construct a statistical test to demonstrate that our local interpolating measure follows a Gaussian distribution, while FedWad IM would reveal statistical information. In addition, we visualize the reconstruction attack towards TriangleWad in lower right panel, where the $\gamma$ follows the gamma distribution. Both local IM and $\tilde{D}_{\text{attack}}$ are uninformative noises.

## 5.2 MEASURE DOCUMENT SIMILARITY WITH PRIVACY

**Datasets** We utilize BBC data processed by Jiang et al. (2023), and use the Word2Vec model Mikolov et al. (2013) to map raw data into embeddings $\mathbf{e}(\cdot)$. We remove stop words, which are generally category independent.

**Baselines** We compare *FedWad* Rakotomamonjy et al. (2024), as it is the only approach that is fit to this case.

| | DirectWad | | | FedWad | | | TriangleWad( 100/1000) | | |
|---|---|---|---|---|---|---|---|---|---|
| **CIFAR10** | 100 | 500 | 1000 | 100 | 500 | 1000 | 100 | 500 | 1000 |
| $\mathcal{W}_2(\mu,\nu)$ | 27.46 | 24.73 | 24.16 | 32.90 | 30.75 | 30.72 | 27.51/32.88 | 24.73/30.76 | 24.16/30.69 |
| $\mathcal{W}_2(D_1^{\text{noise}},\nu)$ | 571.73 | 216.54 | 141.68 | 571.99 | 217.50 | 143.08 | 571.74/572.01 | 216.54/217.02 | 141.68/143.68 |
| $\mathcal{W}_2(D_2^{\text{noise}},\nu)$ | 975.79 | 376.65 | 248.32 | 975.94 | 377.15 | 249.16 | 975.80/975.88 | 376.65/377.32 | 248.32/249.10 |
| Avg.Gap | - | - | - | 1.95 | 2.49 | 2.93 | **0.05** | **0.00** | **0.00** |
| Avg.time(s) | - | - | - | 2.55 | 34.23 | 92.46 | **0.17** | **1.38** | **3.26** |
| **Fashion** | 100 | 500 | 1000 | 100 | 500 | 1000 | 100 | 500 | 1000 |
| $\mathcal{W}_2(\mu,\nu)$ | 12.68 | 10.94 | 10.45 | 15.59 | 14.30 | 15.22 | 12.68/15.67 | 10.94/15.77 | 10.45/15.34 |
| $\mathcal{W}_2(D_1^{\text{noise}},\nu)$ | 295.17 | 107.62 | 70.51 | 295.29 | 108.14 | 71.81 | 295.17/296.38 | 107.62/107.71 | 70.51/71.88 |
| $\mathcal{W}_2(D_2^{\text{noise}},\nu)$ | 687.70 | 269.77 | 178.21 | 687.76 | 270.06 | 179.22 | 687.70/688.40 | 269.78/270.88 | 178.21/179.17 |
| Avg.Gap | - | - | - | 1.02 | 1.39 | 2.35 | **0.00** | **0.00** | **0.00** |
| Avg.time(s) | - | - | - | 1.76 | 18.62 | 54.72 | **0.08** | **0.69** | **3.32** |
| **MNIST** | 100 | 500 | 1000 | 100 | 500 | 1000 | 100 | 500 | 1000 |
| $\mathcal{W}_2(\mu,\nu)$ | 15.05 | 12.99 | 12.57 | 18.66 | 17.13 | 17.04 | 15.05/18.40 | 12.99/17.02 | 12.57/17.66 |
| $\mathcal{W}_2(D_1^{\text{noise}},\nu)$ | 290.19 | 114.30 | 75.81 | 290.37 | 115.12 | 76.90 | 290.19/290.88 | 114.30/115.13 | 75.81/76.88 |
| $\mathcal{W}_2(D_2^{\text{noise}},\nu)$ | 688.94 | 274.37 | 181.39 | 688.98 | 275.06 | 182.04 | 688.94/689.69 | 274.37/275.54 | 181.39/ 182.88 |
| Avg.Gap | - | - | - | 1.27 | 1.88 | 2.07 | **0.00** | **0.00** | **0.00** |
| Avg.time(s) | - | - | - | 1.44 | 17.22 | 65.33 | **0.07** | **2.44** | **9.71** |

Table 1: Quantitative Comparisons in the balanced OT problem: DirectWad represents the ground-truth, we compare FedWad and TriangleWad on the approximation error and computational time.

| **Raw Data** | **FedWad** | **TriangleWad** |
|---|---|---|
| cabinet anger at brown cash raid ministers are unhappy about plans to use whitehall cash to keep council tax bills down  local government minister nick raynsford has acknowledged.  gordon brown reallocated 512m from central to local government budgets in his pre-budget report on thursday. mr raynsford said he had held some  pretty frank discussions with fellow ministers over the plans. | Brown unhappy minister cash council Gordon government Budgets council tax | Volkswagen_Passats_amid Kapersky_Labs Neiman_Marcus_Saks PM_Boyko_Borisov Designer_Outlets Fiuczynski Imperial_College broadcaster_Tommy_Heinsohn Jones occasional_flashes |
| low-budget film wins cesar a film that follows a group of alienated youth in a paris suburb as they prepare to perform an 18th century play has won france s top cinema award. | africa travails paris cinema featuring for french actor at wistful wanted show best for for an the cesar youth | By_DAVID_KINVIG awards grommet foul_mouthed_misanthrope Darr SuperNova_Acceleration_Probe unopened_envelopes_relating sara hero schoolteacher M5_Motorway Spotlighting |

Figure 3: BBC Data: words in highlight are words retrieved by $\mathbf{e}(\mu)$. We randomly choose words retrieved by embeddings of FedWad $\mathbf{e}(\xi^{(K)})$ and embeddings of TriangleWad $\mathbf{e}(\eta_\mu)$.

This experiment aims to demonstrate that using interpolating measures in FedWad raises more serious privacy concerns for text data compared to image data. In images, the interpolating measures provide a visual recognition of each image, but not the original data statistics. However, with text data, the interpolating measures can accurately retrieve original words of raw data in embedding space, causing privacy leakage. Specifically, after computing the Wasserstein distance, we employ the `similar_by_vector` function to explore the most similar words with $\mathbf{e}(\xi^{(K)})_i$ and $\mathbf{e}(\eta_\mu)_i$ respectively. In Figure 3, we observe the text retrieved from $\mathbf{e}(\xi^{(K)})$ matches words in the raw data $\mu$, but for $\mathbf{e}(\eta_\mu)$, most words are unrelated. We define the *matching rate* as the proportion of words retrieved by $\mathbf{e}(\cdot)$ that are identical to the words in the original text, e.g.words retrieved by $\mathbf{e}(\mu)$. When comparing two different texts $\mathcal{W}_2(\mathbf{e}(\mu),\mathbf{e}(\nu))$, the matching rates for $\xi^{(K)}$ and $\eta_\mu$ are 69% and 4%.

## 6 CONCLUSION

In summary, we introduce TriangleWad, a novel approach to efficiently and effectively compute the Wasserstein distance among datasets stored by different parties. We provide a detailed analysis of the approximation bound and the privacy benefits of our proposed approach, along with empirical results demonstrating its practical effectiveness through simulations on various problems, such as data valuation in FL and data selection in data markets. Extensive experiments showcase its superior performance across various tasks involving image and text data.

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

# A  PROOF OF THEOREM 1

Our approach mainly focus on the discrete OT problem. However, we also provide the proof for the general continuous OT problem, where we have a similar conclusion: For 2-Wasserstein distance, the approximation error is bounded by the variance of $\gamma$ is it follows the Gaussian distribution.

## A.1  PROOF OF THE DISCRETE OT PROBLEM

Before proving the Theorem 1, we provide the essential property 1 from Panaretos & Zemel (2019) as follows

**Property 1** *For any vector $x \in \mathbb{R}^{d \times 1}$, $\mathcal{W}_2(X + x, Y + x) = \mathcal{W}_2(X, Y)$.*

We will begin our proof with the case of Gaussian distributions, as their Wasserstein distance has a clear analytical form, which could provide a rigorous approximation error bound. However, our theoretical analysis can be extended to more complex distributions.

Suppose $X_a \in \mathbb{R}^{m \times d} \sim \mathcal{N}(\mu_a, \sigma_a^2)$, $X_b \in \mathbb{R}^{n \times d} \sim \mathcal{N}(\mu_b, \sigma_b^2)$, $\gamma \in \mathbb{R}^{k \times d} \sim \mathcal{N}(\mu_\gamma, \sigma_\gamma^2)$. We consider 2-Wasserstein distance and the Kantorovich relaxation of mass splitting. Without loss of generality, we set $t = 0.5$. Then based on the barycentric mapping, the interpolating measures are

$$\eta_{X_a} = 0.5 \times X_a + 0.5 \times m[\pi(X_a, \gamma)\gamma],$$
$$\eta_{X_b} = 0.5 \times X_b + 0.5 \times n[\pi(X_b, \gamma)\gamma], \tag{15}$$

where $\pi(X_a, \gamma) \in \mathbb{R}^{m \times k}, \pi(X_b, \gamma) \in \mathbb{R}^{n \times k}$ are optimal transport plans.

(1) When $k = 1, \gamma = [\gamma_1, \cdots, \gamma_d]_{1 \times d}, \pi(X_a, \gamma) = [\frac{1}{m}]_{m \times 1}, \pi(X_b, \gamma) = [\frac{1}{n}]_{n \times 1}$, then based on Property 1, $2\mathcal{W}_2(\eta_{X_a}, \eta_{X_b}) = \mathcal{W}_2(X_a + \gamma, X_b + \gamma) = \mathcal{W}_2(X_a, X_b)$

(2) When $k > 1$ and $k \neq m \neq n$. $\pi(X_a, \gamma) \in \mathbb{R}^{m \times k}, \pi(X_b, \gamma) \in \mathbb{R}^{n \times k}$. For $\pi(X_a, \gamma)$, we define $w_{i,l}$ as the value of the $(i, l)$-position value, where $i \in [1, m], l \in [1, d], w_i = \sum_{l=1}^{d} w_{i,l} = \frac{1}{m}$. Further, with uniform weights, there are $\lfloor \frac{m+k-1}{m} \rceil$ non zero elements in each row of $\pi(X_a, \gamma)$. We denote the indices of the nonzero values in each row as the set $\mathcal{I}_i$. For simplicity, we assume all non-zero elements in $\pi(X_a, \gamma)$ has an uniform weight of $\frac{1}{m+k-1}$.

a. $k \to \infty$, then the weight is around $\frac{1}{k}$ if $l \in \mathcal{I}_i$ and 0 otherwise. In geometirc view, each point in $X_a$ are splited to map $k$ points in $\gamma$. Then we have

$$2\eta_{X_a} = X_a + m \times [\sum_{l=1}^{k} w_{i,l} \times \gamma_{l,j}]_{i,j=1}^{m,d} = \frac{m}{k} \times k[\mathbb{E}(\gamma_1), \cdots, \mathbb{E}(\gamma_d)]$$
$$= m[\bar{\gamma}_1, ..., \bar{\gamma}_d]_{1 \times d} \tag{16}$$

Then based on the Property 1 we have $2\mathcal{W}_2(\eta_{X_a}, \eta_{X_b}) = \mathcal{W}_2(X_a, X_b)$.

b. When $k < \infty$, $2\eta_{X_a} = X_a + m \times [\sum_{l=1}^{k} w_{i,l} \times \gamma_{l,j}]_{i,j=1}^{m,d} = X_a + m \times \frac{1}{m+k-1}[\sum_{l=1}^{k} \mathbb{I}_{l \in \mathcal{I}_i} \gamma_{l,j}] = X_a + m \times \frac{1}{m+k-1} \times \frac{m+k-1}{m}[\bar{\gamma}_{i,j}^a]_{l,j=1}^{m,d} = X_a + [\bar{\gamma}_{i,j}^a]_{l,j=1}^{m,d}$. Similarly, $\eta_{X_b} = X_b + [\bar{\gamma}_{i,j}^b]_{i,j=1}^{n,d}$. If we denote $\bar{\gamma}^a = [\bar{\gamma}_{i,j}^a]_{l,j=1}^{m,d} = [\mu_\gamma + \sigma_a Z_a]$, $\bar{\gamma}^b = [\mu_\gamma + \sigma_b Z_b]$, where $Z_a \in \mathbb{R}^{m \times d} \sim \mathcal{N}(0, 1), Z_b \in \mathbb{R}^{n \times d} \sim \mathcal{N}(0, 1)$, then

$$\sigma_a^2 = Var(\frac{m}{m+k-1} \sum_{l \in \mathcal{I}_i} \gamma_{l,j}) = [\frac{m}{m+k-1}]^2 Var(\sum_l \gamma_{l,j}). \tag{17}$$

As $\gamma_{l,j}$ is i.i.d sampled from $\mathcal{N}(\mu_\gamma, \sigma_\gamma^2)$, then $Var(\sum_l \gamma_{l,j}) = \sum_l Var(\gamma_{l,j}) = \sum_l \sigma_\gamma^2 = \frac{m+k-1}{m} \sigma_\gamma^2$. We can get $\sigma_a^2 = \frac{m}{m+k-1} \sigma_\gamma^2$. Similarly, $\sigma_b^2 = \frac{n}{n+k-1} \sigma_\gamma^2$

We define $p_a = \sqrt{\frac{m}{m+k-1}}, p_b = \sqrt{\frac{n}{n+k-1}}$. Therefore, our approximation is

$$2\mathcal{W}_2^2(\eta_{X_a}, \eta_{X_b}) = \mathcal{W}_2^2(X_a + p_a \sigma_\gamma Z_a, X_b + p_b \sigma_\gamma Z_b)$$
$$= \|\mu_a - \mu_b\|_2^2 + \|(\sigma_a^2 + p_a^2 \sigma_\gamma^2)^{\frac{1}{2}} - (\sigma_b^2 + p_b^2 \sigma_\gamma^2)^{\frac{1}{2}}\|_2^2. \tag{18}$$

Furthermore, we focus on the second term as

$$\|(\sigma_a^2 + p_a^2\sigma_\gamma^2)^{\frac{1}{2}} - (\sigma_b^2 + p_b^2\sigma_\gamma^2)^{\frac{1}{2}}\|_2^2$$

$$= (\sigma_a^2 + p_a^2\sigma_\gamma^2) - (\sigma_b^2 + p_b^2\sigma_\gamma^2) - 2\sqrt{(\sigma_a^2 + p_a^2\sigma_\gamma^2)(\sigma_b^2 + p_b^2\sigma_\gamma^2)}$$

$$= (\sigma_a^2 - \sigma_b^2) + \sigma_\gamma^2(p_a^2 - p_b^2) - 2\underbrace{\sqrt{(\sigma_a\sigma_b)^2 + (\sigma_a p_b \sigma_\gamma)^2 + (p_a \sigma_\gamma \sigma_b)^2 + (p_a p_b \sigma_\gamma^2)^2}}_{K}$$

$$= \|\sigma_a - \sigma_b\|_2^2 + \sigma_\gamma^2(p_a - p_b)^2 + 2\underbrace{(\sigma_a\sigma_b + p_a p_b \sigma_\gamma^2 - K)}_{H}$$

$$< \|\sigma_a - \sigma_b\|_2^2 + \sigma_\gamma^2(p_a - p_b)^2. \tag{19}$$

Then we can have an upper bound as

$$2\mathcal{W}_2^2(\eta_{X_a}, \eta_{X_b}) < \|\mu_a - \mu_b\|_2^2 + \|\sigma_a - \sigma_b\|_2^2 + \sigma_\gamma^2(p_a - p_b)^2 = \mathcal{W}_2^2(X_a, X_b) + \sigma_\gamma^2(p_a - p_b)^2 \tag{20}$$

Reversely,

$$H = \sigma_a\sigma_b + p_a p_b \sigma_\gamma^2 - \sqrt{(\sigma_a\sigma_b)^2 + (\sigma_a p_b \sigma_\gamma)^2 + (p_a \sigma_\gamma \sigma_b)^2 + (p_a p_b \sigma_\gamma^2)^2}$$

$$= \sqrt{(\sigma_a\sigma_b)^2} + \sqrt{(p_a p_b \sigma_\gamma^2)^2} - \sqrt{(\sigma_a\sigma_b)^2 + (\sigma_a p_b \sigma_\gamma)^2 + (p_a \sigma_\gamma \sigma_b)^2 + (p_a p_b \sigma_\gamma^2)^2}$$

$$> \sqrt{(\sigma_a\sigma_b)^2 + (p_a p_b \sigma_\gamma^2)^2} - \sqrt{(\sigma_a\sigma_b)^2 + (\sigma_a p_b \sigma_\gamma)^2 + (p_a \sigma_\gamma \sigma_b)^2 + (p_a p_b \sigma_\gamma^2)^2}$$

$$= \frac{(\sigma_a\sigma_b)^2 + (p_a p_b \sigma_\gamma^2)^2 - [(\sigma_a\sigma_b)^2 + (\sigma_a p_b \sigma_\gamma)^2 + (p_a \sigma_\gamma \sigma_b)^2 + (p_a p_b \sigma_\gamma^2)^2]}{\sqrt{(\sigma_a\sigma_b)^2 + (p_a p_b \sigma_\gamma^2)^2} + \sqrt{(\sigma_a\sigma_b)^2 + (\sigma_a p_b \sigma_\gamma)^2 + (p_a \sigma_\gamma \sigma_b)^2 + (p_a p_b \sigma_\gamma^2)^2}}$$

$$> -\sqrt{\frac{[(\sigma_a p_b \sigma_\gamma)^2 + (p_a \sigma_\gamma \sigma_b)^2]^2}{2(\sigma_a\sigma_b)^2 + 2(p_a p_b \sigma_\gamma^2)^2 + (\sigma_a p_b \sigma_\gamma)^2 + (p_a \sigma_\gamma \sigma_b)^2}} \tag{21}$$

Therefore, we have a lower bound

$$2\mathcal{W}_2^2(\eta_{X_a}, \eta_{X_b}) = \|\mu_a - \mu_b\|_2^2 + \|\sigma_a - \sigma_b\|_2^2 + \sigma_\gamma^2(p_a - p_b)^2 + 2H$$

$$> \mathcal{W}_2^2(X_a, X_b) + \sigma_\gamma^2(p_a - p_b)^2 - 2\underbrace{\sqrt{\frac{[(\sigma_a p_b \sigma_\gamma)^2 + (p_a \sigma_\gamma \sigma_b)^2]^2}{2(\sigma_a\sigma_b)^2 + 2(p_a p_b \sigma_\gamma^2)^2 + (\sigma_a p_b \sigma_\gamma)^2 + (p_a \sigma_\gamma \sigma_b)^2}}}_{M} \tag{22}$$

As for $M$, we will compare the value of the numerator and the denominator as

$$2(\sigma_a\sigma_b)^2 + 2(p_a p_b \sigma_\gamma^2)^2 + (\sigma_a p_b \sigma_\gamma)^2 + (p_a \sigma_\gamma \sigma_b)^2 - [(\sigma_a p_b \sigma_\gamma)^2 + (p_a \sigma_\gamma \sigma_b)^2]^2$$

$$= 2(\sigma_a\sigma_b)^2 + 2(p_a p_b)^2\sigma_\gamma^4 + (p_b^2 + p_a^2)\sigma^2\sigma_\gamma^2 - [(p_b^2 + p_a^2)^2(\sigma_a\sigma_b)^2]\sigma_\gamma^4$$

$$= (\sigma_a\sigma_b)^2[2 - (p_b^2 + p_a^2)^2\sigma_\gamma^4] + 2(p_a p_b)^2\sigma_\gamma^4 + (p_b^2 + p_a^2)\sigma^2\sigma_\gamma^2, \tag{23}$$

then set $\sigma_\gamma^2 \le \sqrt{\frac{2}{p_a^2 + p_b^2}}$ will definitely guarantee $0 < M < 1$. Therefore, the approximation error $|2\mathcal{W}_2^2(\eta_{X_a}, \eta_{X_b}) - \mathcal{W}_2^2(X_a, X_b)|$ is bounded by $\sigma_\gamma^2(p_a - p_b)^2 \ll \sigma_\gamma^2$. When $p_a = p_b$ or $k \to \infty$, we have $2\mathcal{W}_2^2(\eta_{X_a}, \eta_{X_b}) = \mathcal{W}_2^2(X_a, X_b)$.

Overall, the approximation gap is affected only by $\sigma_\gamma$ and $k$. Specifically, given a larger $k$, $(p_a - p_b)^2$ becomes smaller, resulting in a better estimation.

## A.2 PROOF OF THE CONTINUOUS OT PROBLEM

For the continuous OT problem, we obatin the similar analysis result but without the multiplayer

**Theorem 4** *In Wasserstein space, if $\eta_\mu$ and $\eta_\nu$ are approximated by Eq. equation 5 respectively with the same $t$, then the approximation error $|\mathcal{W}_p^p(\eta_\mu, \eta_\nu) - t\mathcal{W}_p^p(\mu, \nu)|$ is bounded by $\sigma_\gamma^p$, which is the $p$-th sample moments of $\gamma$.*

Suppose the OT plan between $\mu$ and $\gamma$ is $\pi_\mu$, and OT plan between $\nu$ and $\gamma$ is $\pi_\nu$.

**Proof 1** *The Wasserstein distance between the interpolation measure $\eta^\mu$ and $\eta^\nu$ can be written as*

$$\mathcal{W}_p^p(\eta^\mu, \eta^\nu) = \int_{\mathcal{X} \times \mathcal{X}} d_p\Big(\eta_i^\mu, \eta_i^\nu\Big) d\pi(\eta_i^\mu, \eta_i^\nu)$$

$$\stackrel{(a)}{=} \int_{\mathcal{X} \times \mathcal{X}} d_p\Big(t \times x_i^\mu + (1-t) \times m(\pi_\mu Q)_i, t \times x_i^\nu + (1-t) \times m(\pi_\nu Q)_i\Big) d\pi(x_i^\mu, x_i^\nu)$$

$$\stackrel{(b)}{=} \int_{\mathcal{X} \times \mathcal{X}} d_p\Big(t \times x_i^\mu + (1-t) \times x_{\mu(i)}^\gamma, t \times x_i^\nu + (1-t) \times x_{\nu(i)}^\gamma\Big) d\pi(x_i^\mu, x_i^\nu)$$

$$= \int_{\mathcal{X} \times \mathcal{X}} \|t \times x_i^\mu - t \times x_i^\nu + (1-t) \times x_{\mu(i)}^\gamma - (1-t) \times x_{\nu(i)}^\gamma\|^p d\pi(x_i^\mu, x_i^\nu), \tag{24}$$

*where the equation $(a)$ is based on the definition of Wasserstein distance, and $(b)$ comes from the fact that $\pi_u$ is a permutation matrix and for each row $i$, $\pi_u(i, j)$ is non-zero only for $\mu(i)$ column and $\pi_u(i, \mu(i)) = \frac{1}{n}$.*

*According to the triangle inequality and equation 24, we have*

$$\mathcal{W}_p^p(\eta^\mu, \eta^\nu) \le \int_{\mathcal{X} \times \mathcal{X}} \Big\{\|t \times x_i^\mu - t \times x_i^\nu\|^p + \|x_{\mu(i)}^\gamma - x_{\nu(i)}^\gamma\|^p\Big\} d\pi(x_i^\mu, x_i^\nu)$$

$$\stackrel{(a)}{=} t\mathcal{W}_p^p(\mu, \nu) + (1-t)^p \int_{\mu(i) \times \nu(i)} \|x_{\mu(i)}^\gamma - x_{\nu(i)}^\gamma\|^p d\mu(i) \times \nu(i)$$

$$\le t\mathcal{W}_p^p(\mu, \nu) + (1-t)^p \int_{\mu(i)} \|x_{\mu(i)}^\gamma - \bar{x}^\gamma\|^p d\mu(i) + (1-t)^p \int_{\nu(i)} \|x_{\nu(i)}^\gamma - \bar{x}^\gamma\|^p d\nu(i)$$

$$= t\mathcal{W}_p^p(\mu, \nu) + 2(1-t)^p \sigma_\gamma^p, \tag{25}$$

*where the last inequality is due to that $\gamma$ has uniform weights of samples, $\sigma_\gamma^p$ denotes the p-th moments of samples, and $\bar{x}^\gamma$ is the central moment. Similar, we have*

$$\mathcal{W}_p^p(\eta^\mu, \eta^\nu) \ge t\mathcal{W}_p^p(\mu, \nu) - (1-t)^p \int_{\mu(i) \times \nu(i)} \|x_{\nu(i)}^\gamma - x_{\mu(i)}^\gamma\|^p d\mu(i) \times \nu(i)$$

$$\ge t\mathcal{W}_p^p(\mu, \nu) + 2(1-t)^p \sigma_\gamma^p. \tag{26}$$

*Therefore, we can conclude that $|\mathcal{W}_p^p(\eta^\mu, \eta^\nu) - t\mathcal{W}_p^p(\mu, \nu)|$ is bounded by $\sigma_\gamma^p$.*

## B    PROOF OF THEOREM 2

For $\psi \in \Pi(\mu, \gamma, \nu)$, we set

$$\mathcal{W}_2^2 \psi(\eta_\mu(t), \xi) = \int_{\mathcal{X}^3} \|(1-t)x_i + tx_j - x_k\| d\psi(x_i, x_j, x_k) \tag{27}$$

It is clear that $\mathcal{W}_2^2(\eta_\mu(t), \xi) \le \mathcal{W}_2^2 \psi(\eta_\mu(t), \xi)$.

Based on the Hilbertian identity,

$$\|(1-t)x_i + tx_j - x_k\|^2 = (1-t)\|x_i - x_k\|^2 + t\|x_j - x_i\|^2 - t(1-t)\|x_j - x_i\|^2 \tag{28}$$

we have

$$\mathcal{W}_2^2 \psi(\eta_\mu(t), \xi) = (1-t)\mathcal{W}_2^2 \psi(\mu, \xi) + t\mathcal{W}_2^2 \psi(\gamma, \xi) - t(1-t)\mathcal{W}_2^2 \psi(\mu, \gamma) \tag{29}$$

Based on the Proposition 7.3.1 from Ambrosio et al. (2005), there esists a plan $\psi^\dagger$ such that

$$\mathcal{W}_2^2(\eta_\mu(t), \xi) = (1-t)\mathcal{W}_2^2 \psi^\dagger(\mu, \xi) + t\mathcal{W}_2^2 \psi^\dagger(\gamma, \xi) - t(1-t)\mathcal{W}_2^2 \psi^\dagger(\mu, \gamma)$$

$$\ge (1-t)\mathcal{W}_2^2(\mu, \xi) + t\mathcal{W}_2^2(\gamma, \xi) - t(1-t)\mathcal{W}_2^2(\mu, \gamma), \tag{30}$$

which results in the theorem that the Wasserstein space is a positively curved metric space(Theorem 7.3.2 Ambrosio et al. (2005)), thus we have the following relationship

$$\mathcal{W}_2^2(\eta_\mu(t), \xi) \ge (1-t)\mathcal{W}_2^2(\mu, \xi) + t\mathcal{W}_2^2(\gamma, \xi) - t(1-t)\mathcal{W}_2^2(\mu, \gamma), \tag{31}$$

where $\xi$ is the fixed measure. We can then reformulate the right-hand side of equation 31 as follows

$$\mathcal{W}_2^2(\mu, \gamma)t^2 + \big[-\mathcal{W}_2^2(\mu, \xi) + \mathcal{W}_2^2(\gamma, \xi) - \mathcal{W}_2^2(\mu, \gamma)\big]t + \mathcal{W}_2^2(\mu, \xi), \tag{32}$$

which we can find this is a quadratic function with respective to $t$ and each coefficient is a constant.

## C    PROOF OF THEOREM 3

Let $\eta_t := (1-t)x + tx'$. Let $\pi \in \Pi(\mu, \gamma)$ be an optimal transport plan in the sense that

$$\mathcal{W}_2(\mu, \gamma) = \int_{\mathcal{X} \times \mathcal{X}} \|x - x'\|^2 d\pi(x, x') \tag{33}$$

For any $0 \le s \le t \le 1$, define the coupling $\pi_{s,t} := (\eta_\mu(s), \eta_\mu(t))_{\#}\mu \in \Pi_{\omega(s), \omega(t)}$, where $\omega(s) = (\eta_s)_{\#}\mu$ and $\omega(t) = (\eta_t)_{\#}\mu$. Specifically, $\omega(0) = \mu, \omega(1) = \gamma$, then

$$\begin{aligned}
\mathcal{W}_2^2(\omega(s), \omega(t)) &\le \int \|x - x'\| d\pi_{s,t}(x, y) \\
&= \int \|\pi_s(x, x') - \pi_t(x, x')\| d\pi(x, x') \\
&= \int \|((1-s)x + sx') - ((1-t)x + tx')\| d\pi(x, x') \\
&= (t-s)^2 \int \|x - x'\| d\pi(x, x') \\
&= (t-s)^2 \mathcal{W}_2^2(\omega(0), \omega(1)), \tag{34}
\end{aligned}$$

Therefore if $s = 0$, we have proved that

$$\mathcal{W}_2(\mu, \eta_\mu(t)) \le |t - s|\mathcal{W}_2(\mu, \gamma). \tag{35}$$

Then we could leverage the triangle inequality to yield

$$\begin{aligned}
\mathcal{W}_2(\omega(0), \omega(1)) \\
&\overset{(a)}{\le} \mathcal{W}_2(\omega(0), \omega(s)) + \mathcal{W}_2(\omega(s), \omega(t)) + \mathcal{W}_2(\omega(t), \omega(1)) \\
&\overset{(b)}{\le} (s + |t - s| + |1 - t|)\mathcal{W}_2(\omega(0), \omega(1)) \\
&= \mathcal{W}_2(\omega(0), \omega(1)), \tag{36}
\end{aligned}$$

which means (a) and (b) should be equalities. If we dive into

$$\begin{aligned}
\mathcal{W}_2(\omega(0), \omega(s)) + \mathcal{W}_2(\omega(s), \omega(t)) + \mathcal{W}_2(\omega(t), \omega(1)) \\
= (s + |t - s| + |1 - t|)\mathcal{W}_2(\omega(0), \omega(1)) \tag{37}
\end{aligned}$$

we could have the following inequalities based on equation 35

$$\begin{aligned}
\mathcal{W}_2(\omega(0), \omega(s)) &\le s\mathcal{W}_2(\omega(0), \omega(1)) \\
\mathcal{W}_2(\omega(t), \omega(1)) &\le |1 - t|\mathcal{W}_2(\omega(0), \omega(1)), \tag{38}
\end{aligned}$$

therefore the following inequality holds

$$\mathcal{W}_2(\omega(s), \omega(t)) \ge |t - s|\mathcal{W}_2(\omega(0), \omega(1)). \tag{39}$$

Overall, we have proved

$$\mathcal{W}_2(\omega(s), \omega(t)) = |t - s|\mathcal{W}_2(\omega(0), \omega(1)), \tag{40}$$

where $\omega(0) = \mu$ and $\omega(1) = \gamma$, thereby when $s = 0$, we complete the proof.

## D    ADDITIONAL EXPERIMENTS

### D.1    DISTRIBUTIONAL ATTACK RESULTS

The empirical results are shown in Figure 4 (a) and (b). For CIFAR10 data, the left side two plots are $\xi^{(K)}$ in FedWad, visually we observe it is a kind of combination of two pictures. The right-side are our constructed attack data, and we successfully extract raw "cat" and "car" elements within $\xi^{(K)}$, which are originally from $\mu$ and $\nu$. We also visualize more results in Figure 5.

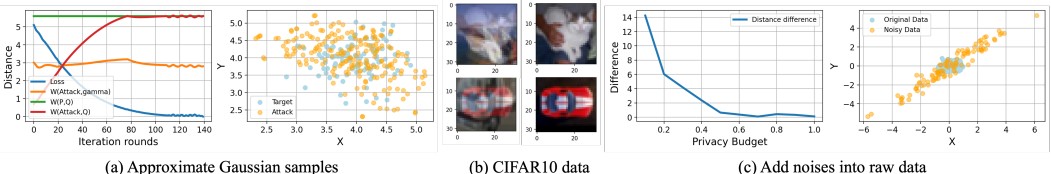

(a) Approximate Gaussian samples  (b) CIFAR10 data  (c) Add noises into raw data

Figure 4: **Attack results (a,b)**: the attack data will gradually converge to the target data with identical distribution; **DP results (c)**: The difference $|\mathcal{W}_2(\mu, \nu) - \mathcal{W}_2(\mu_{\text{perturb}}, \nu)|$ on 2-dimensional Gaussian data results in different level of distance gap with different privacy budget.

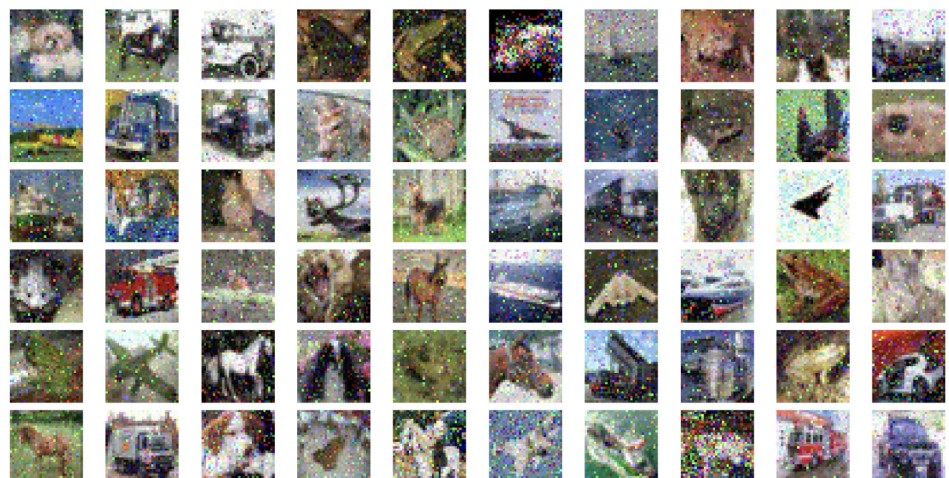

Figure 5: More results on distributional attack

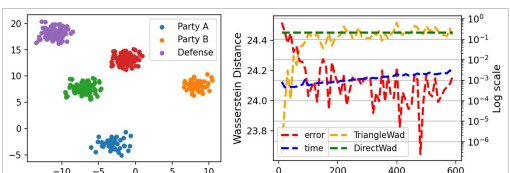

Figure 6: Toy Example

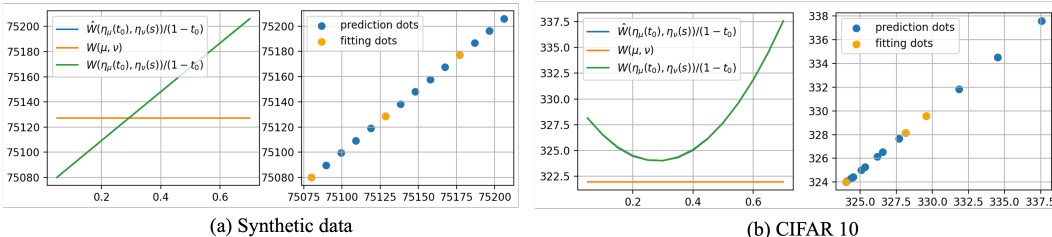

(a) Synthetic data  (b) CIFAR 10

Figure 7: **Line plots:** The lines of Predicted Wasserstein distance (blue) and actual Wasserstein distance (green) between interpolating measures are overlapping. When $t = s_0$, the $\hat{\mathcal{W}}_2(\mu, \nu)$ has minimal gap with $\mathcal{W}_2(\mu, \nu)$ ; **Dot plots:** Predicted distance vs. actual distance between two interpolating measures. Orange dots are for fitting and blue dots are for predictions.

## D.2   TOY ANALYSIS

We illustrate how the intuition behind TriangleWad could be applied to calculate the Wasserstein distance between two Gaussian distributions. We sample 200 data points with different means and the same covariance matrix for Party A, Party B and defense data. For computing local interpolating measures, we set $t = 0.5$ for both sides. In Figure 6, left panel shows out how interpolating measures locating between raw data distributions, and right panel shows the approxited Wasserstein distance and exact one, with different support size for $\gamma$. We set the log value for approximation error and time. The support size does not affect accurations significantly.

## D.3   PREDICTING PERFORMANCE FOR UNKNOWN $t$

In this section, we consider measuring the Wasserstein distance among three data distributions $\mu, \nu_1$ and $\nu_2$ without revealing the value of push-forward parameters. We want to calculate $\mathcal{W}_2(\mu, \nu_1 + \nu_2)$, as mentioned in Sec 3.4. For synthetic data, we consider the balanced OT problem, where $\nu_1 = \sum_{i=1}^{250} x_i^3, x_i \sim \mathcal{N}(12, 10^2), \nu_2 = \sum_{i=1}^{250} x_i^2, x_i \sim \mathcal{N}(3, 1)$ and $\mu = \sum_{i=1}^{500} x_i', x_i' \sim \mathcal{N}(20, 30^2)$. For the CIFAR10 data, we consider unbalanced OT problem, where $\nu_1 = \{x_i, y_i\}_{i=1}^{100}, \nu_2 = \{x_j, y_j\}_{j=1}^{100}, \mu = \{x_i', y_i'\}_{i=1}^{150}$. The labeled dataset is transformed into the vectorial form as discussed before. For simplicity, we define $\nu = \nu_1 + \nu_2$.

We set $t_0 = 0.3$ and sampling ratios are $s_j \in \{0.1, 0.35, 0.60\}$. The randomly initialized $\gamma$ has a standard deviation $\sigma(\gamma) = 3$. We then use the tuple $\{s_j, \mathcal{W}_2(\eta_\mu(t_0), \eta_\nu(s_j))\}$ to fit the function $f(s)$ as described in equation 10, where $\mathcal{W}_2(\eta_\mu(t_0), \eta_\nu(s_j))$ is calculated based on the optimization result with the input of the constructed cost matrix $\mathbf{C}(s_j) = [\mathbf{C}_1(s_j), \mathbf{C}_2(s_j)]^T$. As observed in Figure 7, the predicted values $\frac{1}{1-t_0} \hat{\mathcal{W}}_2(\eta_\mu(t_0), \eta_\nu(s))$ (blue line) and the true values $\frac{1}{1-t_0} \mathcal{W}_2(\eta_\mu(t_0), \eta_\nu(s))$ (green line) are overlapping, which represents our method have a strong representation power. Specifically, we find when $s = t_0 = 0.3$, the green line has an interaction with the true distance $\mathcal{W}_2(\mu, \nu)$ for the synthetic data, or has the minimal gap with the true distance for the CIFAR10 data. It is worthy to note that only $\eta_\mu(t_0), \mathcal{W}_2(\eta_\mu(t_0), \eta_\nu(s_j)), s_j \in \{0.1, 0.35, 0.60\}$ are public information, while $t_0, \eta_\nu(s_j) = \eta_{\nu_1}(s_j) + \eta_{\nu_2}(s_j)$ are kept private.

## D.4   EXPERIMENTS ON THE UNBALANCED OT PROBLEM

We consider two IID and two non-IID cases. Data size is $n_a = 80$, $n_b = 200$. For FedWad, set $n_\xi = m_a + m_b - 1, \xi^{(0)} \sim \mathcal{N}(0, 1)$ and iterations $K = 50$. For TriangleWad, set $n_\gamma = n_\xi, \gamma \sim \mathcal{N}(0, 2)$ (left) or $\mathcal{N}(0, 5)$ (right). For all cases, we set dimension $d_a = d_b = d_\xi = d_\gamma = d = 100$ except the second case we also add $d = 400$. FedWad could not guarantee to find the interpolating in high-dimensional case. The result is shown in Table 2. Additionally, we find the $\sigma(\gamma)$ will affect the approximation error. When the variance becomes larger, TriangleWad provides the approximation with larger difference with the true Wasserstein distance.

| | $\mathcal{N}(10,3)$ | $\mathcal{N}(10,20)$ (d=100,400) | | $\mathcal{N}(10,2),\mathcal{N}(50,3)$ | $\mathcal{N}(100,20),\mathcal{N}(50,30)$ | |
|---|---|---|---|---|---|---|
| DirectWad | 37.27 | 249.10 | 532.46 | 401.47 | 593.20 | |
| FedWad | 41.02 | 276.83 | 575.89 | 401.66 | 607.03 | |
| TriangleWad | 40.99 | 45.73 | 252.32 | 266.51 | 536.05 | 544.05 | 401.78 | 404.01 | 594.91 | 602.90 |

Table 2: **Quantitative comparisons of unbalanced OT problem**: We calculate the Wasserstein distance between two distributions with varying mean, variance and data size. The first two cases calculate the wasserstein distance with the same data distribution; the last two cases calculate the wasserstein distance with the different data distributions

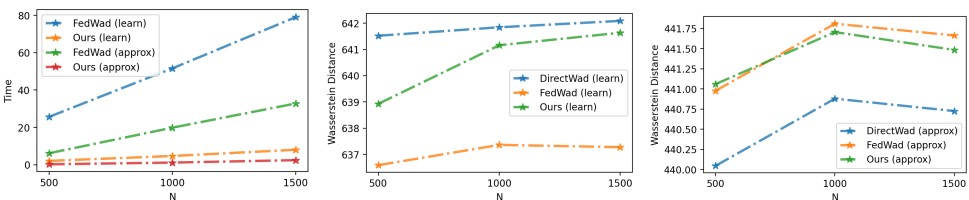

Figure 8: Comparisons when the interpolating measure is exactly calculated/approximated: in both settings, TriangleWad is faster and more accurate than FedWad.

### D.5 ABLATION STUDY

Our approach is also fast and accurate when exactly calculating the interpolating measures instead of approximating. We set $\mu \sim \mathcal{N}(20,5^2), |\mu| = N - 200$ and $\nu \sim \mathcal{N}(100,10^2), |\nu| = N + 200$. Data dimension is set to be $d = 50$. For fair comparisons, we set the supporting size of $\xi^{(0)}$ for FedWad and $\gamma$ for ours as $|\nu| + |\mu| - 1$. The global iteration rounds for FedWad are set to be 10. The experimental results are shown in Figure 8. These three plots show the calculation time and approximated distance of FedWad and TriangleWad when $N = [500, 1000, 1500]$ and $\sigma(\gamma) = \sigma(\xi^{(0)}) = 10$. Our approach is efficient since we only need 3 OT plans in total, thus preventing the computational overhead mentioned in FedWad. Additionally, TriangleWad does not have the significant gap when calculating the exact interpolating, whereas FedWad is unstable and has a larger approximation gap.

## E BROADER APPLICATIONS

### E.1 TRIANGLEWAD OTDD RESULTS

We replicate the experiment of Alvarez-Melis & Fusi (2020) and utilize the code from Rakotoma-monjy et al. (2024) on the labeled data. We conduct the toy example of generating isotropic Gaussian blobs for clustering. We simulate two datasets $D_a = \{\mathbf{x}_a^i, y_a^i\}_{i=1}^{N_a}, D_b = \{\mathbf{x}_b^j, y_b^j\}_{j=1}^{N_b}$ Specifically, we set the data dimension to be $d = 2$. The size of source data to be $N_a = 500$, and the size of target data to be $N_b = 600$. The number of classes is set to be 3. We conduct the data augmentations with the corresponding class-conditional mean and vectorized covariance. To reduce the dimension of the augmented representation, we consider the diagonal of the covariance matrix. Then we calculate the 2-Wasserstein distance with TriangleWad and exact OTDD. The technical details of our approach is as follows: firstly, we construct a matrix $\mathbf{X}_a = [\mathbf{x}_a, m_{y_a}, vec(\Sigma_{y_a}^{1/2})]$. Therefore, we get $\mathbf{X}_a \in \mathbb{R}^{N_a \times (d+d_0)}$, where $d_0$ is the dimension of the class-conditional mean and vectorized covariance. Secondly, we randomly initialize $\gamma \in \mathbb{R}^{k \times (d+d_0)}, k = \min\{N_a, N_b\}$, and construct the interpolating measure $\eta_a(t) \in \mathbb{R}^{N_a \times (d+d_0)}$ with the barycentric mapping. Similarly, $\mathbf{X}_a = [\mathbf{x}_b, m_{y_b}, vec(\Sigma_{y_b}^{1/2})]$ and $\eta_b(t) \in \mathbb{R}^{N_b \times (d+d_0)}$. Finally, we calculate $\hat{\mathcal{W}}_2(\mathbf{X}_a, \mathbf{X}_b) = \frac{1}{1-t}\mathcal{W}_2(\eta_a(t), \eta_b(t))$. This procedure is different to OTDD, where the cost matrix is changed as $d(z, z')$ in equation 12. The data is visualized in Figure 9. In our results, the distance calculated by OTDD is 208.23 and TriangleWad has the result of 210.18. While TriangleWad could have relatively accurate approximation with the augmented form, there is an issue when some data points are mislabeled. For the mislabeled part, it is very important to break the constraint of vectorial representations. We will leave it for the future work.

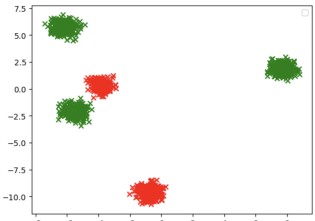

Figure 9: The visualization of synthetic labeled data

| # Clients | ExactFed | GTG | MR | DataSV | FedBary (1000/5000) | TriangleWad |
|---|---|---|---|---|---|---|
| 5 | 31m | 33s | 5m | 25m | 7m / 20m | 2m |
| 10 | 3h20m | 7m | 40m | 2h30m | 14m / 40m | 4m |
| 50 | - | - | - | - | 1h10m / 3h20m | 20m |
| 100 | - | - | - | - | 2h20m / 6h40m | 40m |

Table 3: Evaluation time with different size of $N$: For ExactFed, GTG and MR, we only consider the evaluation time after model training; Evaluation time of FedBary and TriangleWad increase linearly with $N$. A smaller support size in FedBary results in less time, yet a larger distance gap.

### E.2 CONTRIBUTION EVALUATION IN FL

**Datasets** We use all image datasets mentioned before, and follow the same data settings in Liu et al. (2022): We simulate $N = 5$ parties and consider both iid and non-iid cases.

**Baselines** We consider 7 different baselines, in which all of them evaluate client contribution in FL: exact calculation *exactFed*, accelerated *GTG-Shapley* with its variants (*GTG-Ti/GTG-Tib*) Liu et al. (2022), *MR* and *OR* Song et al. (2019), *DataSV* Ghorbani & Zou (2019) and *FedBary* Li et al. (2024b).

We consider exactFed as the ground truth since it precisely calculates the marginal contribution of adding model parameters from one party by considering all subsets, for example, $2^N$ for $N$ parties. In previous quantitative comparisons, we found that the Wasserstein distances computed by Fedwad and our method have trivial differences with Gaussian noises. Shapley-based approaches provide marginal contributions, thus illustrating proportional contributions. On the other hand, FedBary and TriangleWad provide absolute values, so we normalize them to ensure all values fall within the range $[0, 1]$. Overall, Wasserstein-based approaches offer distributional views with correct contribution topology. For case (1), all client contributions are identical due to identical distributions. Case (2) and Case (3) resemble exactFed, while others are more sensitive. Due to computational methods, our approach is more sensitive to features, resulting in wider differences in contribution levels. MR and our method follow the same topology as exactFed, whereas other approximated approaches are completely wrong in this case, e.g. Party 5, with the most noise, has the highest contribution score. We also present the evaluation time of various algorithms. Our approach provide a linear complexity w.r.t to the number of clients, as the evaluation of each client is independent.

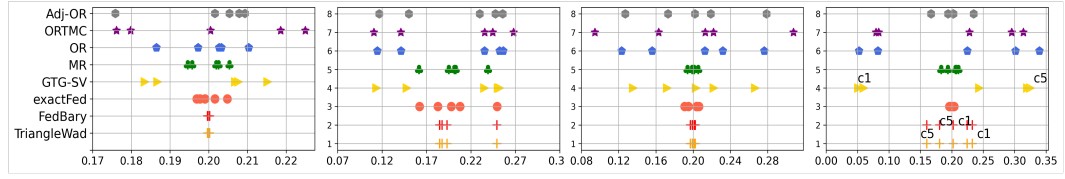

Figure 10: Contribution evaluation of 5 parties with CIFAR10 data

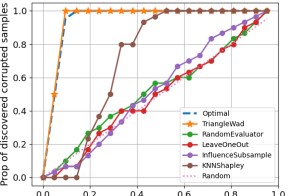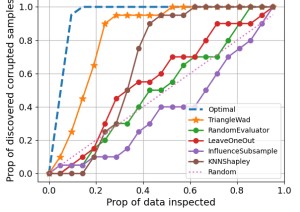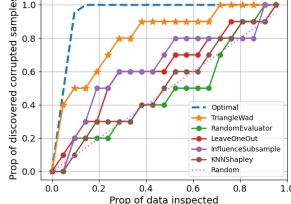

Figure 11: Noisy Feature Detection on CIFAR10 and two tabular datasets: Adult and Stock. Our approach has better noisy detection ability compared to other data valuation approaches. It is worthy to note that others need to use raw data, while TriangleWad could be used in the private setting.

### E.3 NOISY DATA DETECTION

We conduct experimental results on one image dataset: CIFAR10 and three tabular datasets: Adult Income, Stock prediction, Fraud detection. Here we randomly choose the proportion $15\%$ of the training dataset to perturb. For the selected training datapoints to be perturbed, we add Gaussian noises $\mathcal{N}(0, 1)$ to the original features. Then KNNShapley, InfluenceFuction, LeaveOneOut utilize raw data to conduct value detection, while TriangleWad use encrypted data to conduct detection, respectively. In most cases, we control $\sigma(\pi\gamma) = \sigma$ to make fair comparisons.

Results are shown in Figure 11. The x-axis represents the proportion of inspected datapoints, while the y-axis indicates the proportion of discovered noisy samples. Therefore, an effective approach should identify more noisy samples with fewer inspected samples. For each method, we inspect datapoints from the entire training dataset in descending order of their scores, as higher scores indicate greater data value. For ours, we use the negative gradient because it has an inverse relationship compared to others. Our approach significantly outperforms other methods. Notably, even when we set a very large $\sigma(\gamma) = 100$ and $|\gamma| = 20$ to increase $\sigma(\pi\gamma)$ for image data, the first $10\%$ of datapoints identified as noisy by us contain $100\%$ of the noisy feature datapoints. This result demonstrates the high effectiveness and robustness of our approach for image data. In other cases, ours also outperforms the best.

### E.4 DATA VALUATION FOR BOOSTING TEST PERFORMANCE

In the practical data acquisition scenarios, a data buyer has a specific goal and wants to buy training data to predict their test data Lu et al. (2024). Specifically, given a set of unlabeled test data $D_{\text{test}} = \{x_1^{\text{test}}, \cdots, x_m^{\text{test}}\}$, the data valuation and selection task is to select valuable subsets of training data points from the data sellers $D_{\text{train}} = \{(x_j^{\text{train}}, y_j^{\text{train}})\}_{j=1,\cdots,n}$, so that the model trained on these valuable data points will have a smaller prediction loss on the test data. Notably, in this setting, we do not incorporate the labels of the training data.

Following a similar experimental setup as in Lu et al. (2024), we conduct experiments on one synthetic Gaussian dataset and one real-world medical dataset: the RSNA Pediatric Bone Age dataset Halabi et al. (2019), where the task is to assess bone age (in months) from X-ray images. To extract features of RSNA Pediatric Bone Age dataset, each image is embedded using a CLIP ViT-B/32 model Radford et al. (2021). We set $\|D_{\text{train}}\| = 1000$ and $\|D_{\text{test}}\| = 50$, selecting training data under varying selection budgets. Specifically, two interpolating measures $\eta_{x^{\text{train}}}(t)$ and $\eta_{x^{\text{test}}}(t)$ are constructed via equation 5. These measures are then used as inputs to compute the gradient score via equation 13. After optimization, we select the top-$k$ most valuable data points (those with the largest negative gradient scores) and train a regression model to predict the test data. We compare our approach with other baselines on the test mean squared error (MSE). As shown in Figure 12, our data selection algorithm achieves lower prediction MSE on the test data compared to the baselines.

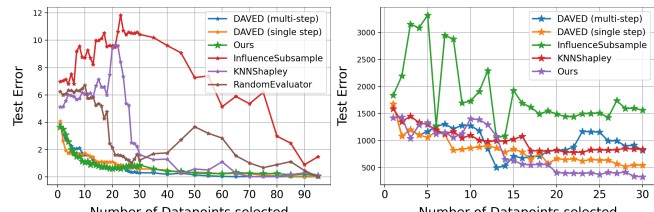

Figure 12: Our approach has low test error (MSE) on both synthetic data and real-world medical imaging data

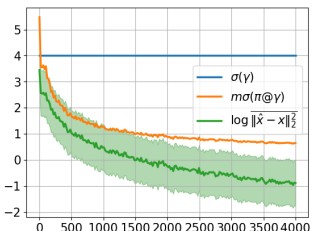

Figure 13: Empirical results of Corollary 2: $\sigma(\pi(\mathbf{x}, \gamma)\gamma$ continuously decreases as $k$ increases. Simultaneously, the logarithmic approximation error drops to negative value, which means $\hat{\mathbf{x}} \to \mathbf{x}$. This result demonstrates how the size and variance of $\gamma$ determine the data privacy level.

## F   EMPIRICAL RESULTS FOR THEORETICAL ANALYSIS

### F.1   EMPIRICAL RESULTS OF COROLLARY 2

This section validates the theoretical analysis of Corollary 2. Without loss of generality, we set $\mathbf{x} \in \mathbb{R}^{m \times d} \sim \mathcal{N}(0, 1), m = 100, \gamma \in \mathbb{R}^{k \times d} \sim \mathcal{N}(0, 16)$, where $k$ is increased from 1 to 4000. Based on equation 9, we approximate $\hat{\mathbf{x}} = \frac{1}{1-t}\left(\eta_{\mathbf{x}}(t) - t \times \bar{\gamma} - t \times \sigma(\pi^{\star}(\mathbf{x}, \gamma)\gamma)\right)$, and calculate the average approximation loss as $\overline{\|\hat{\mathbf{x}} - \mathbf{x}\|_2^2}$. To better visualize the values, we apply a logarithmic scale $\log(\cdot)$ for the approximation loss. The experimental results are shown in Figure 13. We observe that the approximation loss (green line) continuously decreases as $k$ increases. Additionally, the standard deviation (orange line) converges to a very small value.

### F.2   QUANTIFY THE DISSIMILARITY OF RAW DATA AND INTERPOLATING MEASURE

This section validates the theoretical analysis of Theorem 3. We conduct the experiments on the CIFAR10 data set. We calculate the Wasserstein distance $\mathcal{W}_2(\eta_\mu(t), \eta_\nu(t))$ when $t$ increases from 0.1 to 0.9. The result is shown in Table 4. The groundtruth distance is 806.4. We could find our approximation serves the robustness with the relatively large push-forward value $t$, due to the geometric property. However, in general perturbations, $\mathcal{W}_2(\mu + \gamma, \gamma) = 10.9$ when $\sigma(\gamma) = 1$. When $t$ becomes 0.9, there might be a large deviation as the interpolating measure is very close to the random gaussian distribution $\gamma$. Overall, we can set a large value of $t$ to increase the dissimilarity of raw data and interpolating measure, to protect the privacy.

| $\sigma(\gamma)$ | $t$ | 0.1 | 0.2 | 0.3 | 0.4 | 0.5 | 0.6 | 0.7 | 0.8 | 0.9 | groundtruth |
|---|---|---|---|---|---|---|---|---|---|---|---|
| 1 | $\mathcal{W}_2(\eta_\mu, \eta_\nu)$ | 806.5 | 806.5 | 806.6 | 806.6 | 806.8 | 807.0 | 807.4 | 807.8 | 808.7 | 806.4 |
| | $\mathcal{W}_2(\eta_\mu, \mu)$ | 20.54 | 41.09 | 61.6 | 82.2 | 102.7 | 123.3 | 143.8 | 164.4 | 184.9 | - |
| 5 | $\mathcal{W}_2(\eta_\mu, \eta_\nu)$ | 806.3 | 806.6 | 806.9 | 807.3 | 807.5 | 807.9 | 808.9 | 811.8 | 818.9 | 806.4 |
| | $\mathcal{W}_2(\eta_\mu, \mu)$ | 22.3 | 44.7 | 67.1 | 89.4 | 111.8 | 134.2 | 156.6 | 178.9 | 201.3 | - |

Table 4: Given fixed $\gamma$, we change the parameter $t$ from 0.1 to 0.9. Geometrically, when $t = 0$, $\eta_\mu = \mu$, when $t \to 1$, $\eta_\mu$ is closer to $\gamma$. Although $\eta_\mu$ has different distribution with $\mu$, we could still provide accurate estimation.

