# OpenReview forum: "Private Wasserstein Distance"
_ICLR.cc/2025/Conference — Submitted to ICLR 2025_

### Official Review · Reviewer_UvYZ · 2024-10-30

**Soundness:** 2
**Presentation:** 2
**Contribution:** 2
**Rating:** 3
**Confidence:** 4

**Summary:**

This submission proposes an algorithm called TriangleWad for computing the 2-Wasserstein distance between datasets owned by different parties. The algorithm is based on an observation that the distance between two datasets mu and nu can be estimated by computing the distances between an interpolation of mu to gaussian noise and an interpolation between nu to gaussian noise.

Algorithms for various applications can be based off of this idea, but they all involve a crucial step where Party 1 computes an interpolation between their data and random noise and shares this interpolation with others. The random noise is known to everyone but the interpolation parameter is not.

**Strengths:**

The paper includes interesting analysis of the relation between the wasserstein distances between two datasets as a function of the interpolation measures between the datasets and shared random noise.

Recognizing that public knowledge of the shared noise and interpolation parameter would lead to  a very insecure algorithm, the paper analyzes how to keep the interpolation parameter secret.

**Weaknesses:**

The major weakness in this paper is that there is no formal privacy model and really no provable privacy guarantees. As far as I can infer, the intended privacy model is whether one party's data can be almost completely reconstructed. For the intended applications of federated learning or data market quality checks, that may be a very weak guarantee, as partial reconstruction of some records may be a significant breach (like in a medical database). Since one has to share an interpolating distribution (line 290), the choice of interpolated data points and coefficients (Eq 4) may leak significant information (for example it can be used to estimate t). To pass the bar, this paper needs a privacy model that is appropriate to its intended applications.

Another weakness is the paper is incredibly difficult to read. Some parts are notation-dense while other parts are handwavy. The notation is not always used consistently and not always defined. As a result, I may have not entirely understood everything. Examples include:
- #pi in Line 108
- Theorem 1 is not self-contained (for instance it doesn't define gamma or introduce sigma, so you have to keep looking elsewhere), the dependence of C on k is omitted in the big-O notation and is replaced by the vague non-quantitative "is negatively related to"
- The paper keeps switching between W, W_p, W^2 in an inconsistent use of notation. As p=2, there is no generality afforded by using p in the notation anywhere in the paper.
- Eq 9 sigma(pi*....) does not appear to be defined.
- Is corollary 2 missing text? Is it supposed to start as "Suppose each element of eta(mu) ..." or is there some other meaning to it?
- Corollary 1 appears to contradict the preceding discussion. First,  "hat{W}_p(mu, nu) is an unbiased estimation " of what? of W(mu, nu)? How does it square with Eq 7 saying that it is as least as large (which, when combined with unbiasedness should mean that they are identical)?

**Questions:**

1. What is the formal privacy model in this paper? I don't think one can claim privacy without one.

2. What are all of the possible attack vectors that can be used? Are the ones listed in the weaknesses correct?

3. Why is the privacy model an appropriate one for the various applications that were listed?

4. What formal privacy guarantees can you provide with respect to this model?

---

### Official Review · Reviewer_5mo1 · 2024-10-31

**Soundness:** 3
**Presentation:** 2
**Contribution:** 4
**Rating:** 6
**Confidence:** 3

**Summary:**

The authors propose the Triangle Wasserstein distance, which is a mechanism to calculalte the Wasserstein distance between two measures in a secure way, i.e., while not sharing information that may be used by any other part to recover the data distribution. The method relies on a reference Gaussian measure and a clever geometric argument that allows us to use one distance to calculate the other. The authors show that the existing measure, FedWad is vulnerable to an attack that learns the data distribution, which makes their method well motivated.

TL;DR - I think this is a great contribution that can benefit from a better presentation.

**Strengths:**

- I really liked the proposed technique and I think this is a great contribution. Starting by showing that the existing method is vulnurable to an attack well motivates your work. Then, the proposed scheme is relatively simple, yet strong. I think this work is highly relevant to current topics
 that are keeping our community busy these days.

- I liked the discussion on broader applications, which strnegths this paper.

- I liked the adversarial attack design.

- The authors present experiments in a variety of fields - This work my be highly relevant across multiple fields and can be modified to extend to even more.

**Weaknesses:**

1. In contrary to the contribution, the presentation sometimes feels a little too heuristic. It feels that some statements are not given formally, but are framed within discussions rather than theorems, which makes this paper less formal than it can be. For example (some were transformed into questions):
  a. The theoretical analysis section mainly consists of a discussion (only one theorem). I would expect elements such as complexity analysis, privacy analysis, etc, to be framed within formal statements.

2. While I think the broader applications are important, the authors do not present any specific results on experiment on them. I would therefore expect the discussion to be brief in the main paper, and elaborated on in the appendix.

technical:
1. I am not sure the citation format is good.
2. line 111 - ‘geodesic’ instead of ‘geodesics’
4. line 142 - ‘valuation’ - did you mean evaluation?
5. equation 6 should depepend in 't' somehow, it is not conveyed by the notation in any way.
6. line 188 - ‘6’ → ‘equation 6’
7. 196 - ‘computes’→’compute’
8. paragraph 187-199 is not easy to follow, while being crucial to the overall flow.
9. ‘proof is shown in appendix’ - refer to specific appendecies.
10. corollary 1 should be rewritten, perhaps in the sape of a list of conditions.
11. line 435 - ‘timeand’
12. line 938 - ‘our’ →’out’
13. 958-959- ‘worthy to note’→’noteworthy’

**Questions:**

1 . Theorem 1 - can the authors explicitly mention how does the bound 'negatively depend on k'? do they have a rate?
2. Unbiased estimation (line 250) should be defined.
3. corollary 2 - ' mean degree of linear transformation' should be defined.
4. Theorem 2 - the order of mathematical quantifier is confusing (for all, etc.) - The theorem can be better written.
5. Can the authors elaborate on what can be learned from figure 2 in the attack image? I don't find it clear at the moment.
6. Could the authors elaborate on what can be learned from the ablation study?

Comments (not really weaknesses or questions):
1. In terms of presenting the weakness of FedWad, I feel that the figure you present in the appendix (fig 5) is more convincing than the one in the main paper (fig 2).
2. I would add captions to the appendix figures - they are a little hard to follow at the moment.
3. the paragraph ‘data evaluation in FL an data marketplace’ is hard to follow. I would rewrite it in a way that better conveys what was done.

---

### Official Review · Reviewer_814S · 2024-11-03

**Soundness:** 3
**Presentation:** 2
**Contribution:** 2
**Rating:** 5
**Confidence:** 2

**Summary:**

This paper studies the problem of computing the Wasserstein distance in privacy-preserving manner. A method named TriangleWad is designed, which leverages geometric properties within the Wasserstein space to ensure that raw data remains hidden while maintaining high estimation accuracy.

**Strengths:**

1. The studied problem is well motivated.

2. This proposed method leverages geometric properties within the Wasserstein space, which provides some insights into the studied problem.

3. The paper provides a thorough theoretical analysis of the geometric properties.

**Weaknesses:**

1. It appears that the method relies on certain assumptions, such as the availability of a global shared random distribution and the proportional relationship between interpolating measures. These assumptions might not be readily available in practical settings, potentially limiting the method’s applicability.

2. Although the paper discusses privacy benefits, it is unclear how much privacy can be guaranteed. The authors include some empirical analysis against some attacks. Is it possible to derive the privacy bound?

3. While the paper includes extensive experiments, it could benefit from more practical case studies that illustrate the real-world applicability of TriangleWad.

4. The presentation could be enhanced. The paper is dense with notations and formulas. It would be beneficial if the authors provided more intuitive explanations and practical examples.

**Questions:**

Please refer to the Weaknesses section.

---

### Official Review · Reviewer_EUqC · 2024-11-04

**Soundness:** 3
**Presentation:** 3
**Contribution:** 3
**Rating:** 6
**Confidence:** 4

**Summary:**

This paper proposed a novel algorithm which estimates the Wasserstein distance between two datasets (held by two different parties) without leaking too much information to the other party, such that one party cannot reconstruct the data distribution of the other party. The high level idea of the proposed algorithm is inspired by similar triangles: Let two datasets considered be \mu and \nu, we pick an arbitrary dataset \gamma, if \eta_{\mu}(t) is a linear interpolation between \mu and \gamma (i.e., t * \mu + (1-t) * \gamma), and \eta_{\nu}(t) is defined similarly, then (distance between \eta_{\mu}(t) and \eta_{\nu}(t))/t is a good approximation to the distance between \mu and \nu. The paper also shows that if two parties use different t, how do they get the approximation. In addition, the paper also proves that: given the information, the dataset distribution of one party is impossible to be reconstructed by the other.

**Strengths:**

1. This work can be seen as a big improvement of the previous work (11). In the previous work (11), the drawback is that the middle point will eventually be on the Optimal Transportation path between \mu and \nu. However, in this work, authors cleverly choose an arbitrary middle point and use a similar triangle technique to estimate the distance.

2. Authors theoretically proved that the information leaked by the proposed algorithm is not enough for the other party to reconstruct the data distribution.

3. The entire paper shows solid theoretical analysis.

4. Empirical results are also impressive.

**Weaknesses:**

1. There is no discussion about the privacy guarantee of the algorithm in a traditional view (such as Differential Privacy). In fact, the proposed algorithm is definitely not DP. It would be good to discuss whether there is a potential to convert the algorithm into DP with additional loss in accuracy.

2. The paper mainly focuses on W-2 Wasserstein distance. It would be good to clarify whether the algorithm still holds for other p. In practice, W-1 (i.e., Earth-Mover-Distance) is also important in practice.

3. The algorithm without leaking t is asymmetric, i.e., only one party does not reveal t, but the other party needs to reveal multiple t.

**Questions:**

1. Is it possible to further modify the algorithm to make it DP?

2. Does the algorithm generalize to the general W-p case?

3. Is it possible to further improve the algorithm to not reveal t from either party?

---

### Official Review · Reviewer_ftyh · 2024-11-04

**Soundness:** 2
**Presentation:** 2
**Contribution:** 3
**Rating:** 5
**Confidence:** 2

**Summary:**

The paper discusses a distributed method to estimate the Wasserstein distance between a source and (possibly several) target distributions. This problem finds applications in data valuation, and domain adaptation and selection (among others). The paper considers the data privacy of individual parties involved and describes an attack on the prior work which requires interpolation between target and source distributions. The paper then proposes a method to improve upon this, by instead sharing interpolations with a random gaussian distribution. The proposed method improves on prior work performance while providing a symmetric interface for all parties involved.

**Strengths:**

- Studies relevant problem (data evaluation)
- Empirical results show the proposed method is performant
- Providing symmetric access to measure estimates for all parties involved is an interesting contribution of the work

**Weaknesses:**

- The paper is not well-structured in my humble opinion. For instance, there are gaps between related questions and concerns (3.4 and then again 4.2.1), while a list of interesting but flow-breaking applications appear in the middle Section 3.5.

- Claims of improved privacy are not substantiated very well. The paper recognizes the issues with privacy claims but the arguments provided do not assuage validity concerns (empirical results are not proof of privacy). For instance

	- I think an attack similar to what authors present against FedWad (in Section 3.2)
	  to learn $\mu$ from $\nu$ is possible. In fact, it seems the whole security argument comes from lack of iterations, and not the fundamental design of the proposed solution in Section 3.3

		-
		  > Page 8: TriangleWad does not calculate any interpolating measure between µ and ν, so the attacker can not use available information to identify the distribution of the raw data.


		  This is a pretty weak argument. There are other pieces of shared information that the attacker can use. Including the shared $\gamma$ , and the fact that good performance is only possible when both parties have the same $t$ shows a pretty strong utility-privacy trade-off

	-
	  > 3.4 APPROXIMATE WASSERSTEIN DISTANCE WITH UNKNOWN t

		- The arguments in Section 3.4 are not satisfying in hiding the $t$ parameter. To me the only way to prove such a result, is to show that inverting (10) is not possible (which I have strong doubts on)

		- I do not have a concrete proof, but I believe you can construct seller distributions $\eta_{\nu_i} (s_j)$ such that you can invert the relationship can find $t_0$

- The formal presentation of the paper can be considerably improved. For instance,

	-
	  > Page 5: and it has a negative relationship with k

		- What does this mean in a formal proof?

	- Undefined Notation in $f_l^{\star}$ in eq.13. Lack of proper introduction of the "Monge" problem in the background.

- Empirical section shows improvement over the baseline but needs improvement. In particular,

	- I am skeptical of nearly all zero gaps reported in Table 1.

	- I was expecting a trade-off utility-privacy trade-off (for instance, as a result of the procedure in 3.4) however I do not think the method presented in Section 3.4 is validated in the empirical section at all.

	- None of the results have confidence intervals (or any other measure of uncertainty).

	- The paper relegating important empirical results to the appendix. For instance, the rather surprising result at the end of 4.2.2. (BTW, please include the full paper in the submission)

- Minor

	- The paper uses non-standard citations for ICLR

	- Bad formatting of citation as subject. For example on page 3:
	  > (11 ) proposes

**Questions:**

- Can you provide a stronger argument (hopefully, theoretical in nature) for the distributional attack?

- Can you evaluate the "collusion of sellers" attack vector that I delineated in Weaknesses (in response to Sec. 3.4)?

- As I have discussed, I am fairly skeptical of the privacy claims of the method. Having said that, the paper presents an interesting method and applications by in a rather ad-hoc manner. Can you briefly summarized your methods contribution modulo the privacy claims? Are you willing to weaken those claims? If so, please provide a formal adversary model and the weakened privacy protection claims for us to re-evaluate.

---

### Meta-Review · Area_Chair_TE13 · 2024-12-20

**Metareview:**

The submission addresses the challenge of computing the Wasserstein distance across privacy-sensitive datasets. The authors propose "TriangleWad," an algorithm for computing the Wasserstein that aims to balance privacy and utility. They propose an attack on FedWad (a prior attempt for privacy-preserving Wasserstein distance computation), and argue how TriangleWad addresses the limitations of this method.

The reviewers recognized that the problem is well-motivated, acknowledged that the paper contains promising ideas, and mostly agreed that privacy-ensuring methods for Wasserstein distance computation could be of broader relevance. In particular, reviewer EUqC highlighted that "This work can be seen as a big improvement of the previous work," while Reviewer ftyh (and others) appreciated the effort to address limitations in prior methods However, several concerns emerged during the review process.

Most notably, the reviewers criticized the lack of a formal privacy model and guarantees (ftyh, UvYZ), stating that "this is an interesting work but the contributions are exaggerated through imprecise specification" (ftyh). While the authors argue that raw data reconstruction is infeasible without access to the optimal transport (OT) plan, reviewers remain unconvinced without a formal adversary model or proof. Reviewer 814S raised concerns about practical assumptions (e.g., global shared random distributions), yet did not engage with the rebuttal. Presentation issues, imprecise notation and insufficient clarity in theoretical claims were also flagged (5mo1, UvYZ).

Based on the initial reviews and the fact that most reviewers remained tepid post-rebuttal, as well as my read of the paper, I recommend rejecting the paper at this time. This is an interesting problem, and there is merit in TriangleWad (particularly relative to prior work), but the manuscript needs to be revised before acceptance. I encourage the authors to revise the paper with clearer privacy model, notation, and empirical validation of the algorithm’s robustness against concrete threat models.

**Additional Comments On Reviewer Discussion:**

The reviewers remained tepid post-rebuttal and mostly stood by their scores. In particular, reviewer 814S did not engage with the authors so their review receives less weight.

---

### Decision · Program_Chairs · 2025-01-22

Reject